# Influence Functions for Edge Edits
# in Non-Convex Graph Neural Networks

**Jaeseung Heo**[1], **Kyeongheung Yun**[2], **Seokwon Yoon**[2],
**MoonJeong Park**[1], **Jungseul Ok**[1,2], **Dongwoo Kim**[1,2]
[1]Graduate School of Artificial Intelligence
[2]Department of Computer Science & Engineering
POSTECH, South Korea
{jsheo12304,yuonsinsa,swyoon,mjeongp,jungseul,dongwookim}@postech.ac.kr

## Abstract

Understanding how individual edges influence the behavior of graph neural networks (GNNs) is essential for improving their interpretability and robustness. Graph influence functions have emerged as promising tools to efficiently estimate the effects of edge deletions without retraining. However, existing influence prediction methods rely on strict convexity assumptions, exclusively consider the influence of edge deletions while disregarding edge insertions, and fail to capture changes in message propagation caused by these modifications. In this work, we propose a proximal Bregman response function specifically tailored for GNNs, relaxing the convexity requirement and enabling accurate influence prediction for standard neural network architectures. Furthermore, our method explicitly accounts for message propagation effects and extends influence prediction to both edge deletions and insertions in a principled way. Experiments with real-world datasets demonstrate accurate influence predictions for different characteristics of GNNs. We further demonstrate that the influence function is versatile in applications such as graph rewiring and adversarial attacks.

## 1 Introduction

Graph neural networks (GNNs) have demonstrated that leveraging structural relationships, often encoded as connectivity between data points, can enhance the predictive performance of neural networks across many tasks. Although the literature clearly identifies the importance of the relationship, the individual contribution of each connectivity, i.e., an edge, remains poorly understood.

Several recent studies have explored edge importance from a particular perspective. For example, Nguyen et al. [26] propose an edge rewiring method to mitigate the problem of over-smoothing [23], a phenomenon where the learned node representation becomes indistinguishable as the depth of the GNN increases. Alon and Yahav [2] suggest edge rewiring methods to overcome the over-squashing, which occurs when information propagation encounters bottlenecks between distant nodes.

Despite progress in addressing individual challenges, a unified framework for quantifying edge influence across these perspectives would provide a more comprehensive understanding of their role in graph neural networks. For instance, it would allow us to assess how modifying a single edge affects model behavior from both over-smoothing and over-squashing perspectives. On the other hand, influence functions have been introduced to quantify the impact on evaluation metrics, such as validation loss, when a training data point is removed [21]. To do so, the function estimates the *changes in model parameters* when the target data point is excluded from the training.

39th Conference on Neural Information Processing Systems (NeurIPS 2025).

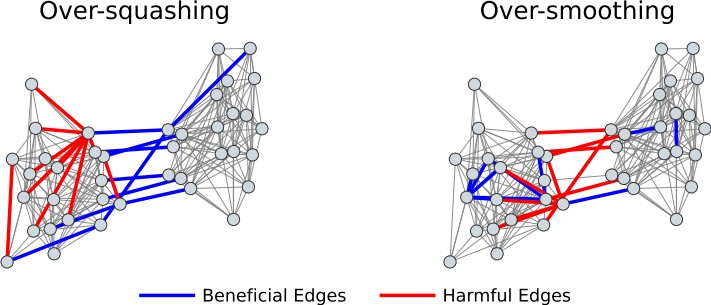

Figure 1: An illustration of beneficial and harmful edges identified by an influence function proposed in this work with respect to two evaluation metrics. A harmful edge is one that either blocks information propagation between nodes (over-squashing [2]) or makes node representations indistinguishable (over-smoothing [23]). The barbell graph consists of two clusters, each with distinct node labels. The influence function can be used to analyze the properties of edges. For example, the edges connecting two different clusters mitigate over-squashing while amplifying over-smoothing.

Applying influence functions [21] to GNNs introduces a unique challenge, primarily because modifying an edge can alter the *propagation paths* and consequently change the underlying *computational graph* structure. Previous attempts to adapt influence functions for GNNs have focused on parameter changes, failing to address the influence of the changes in the computational graph [7, 35]. Moreover, their influence functions rely on the strict convexity assumption of the loss function with respect to the model parameters, which limits their applicability to widely used non-convex GNNs.

In this work, we propose an *influence function* tailored specifically for GNNs, enabling precise predictions regarding the effects of edge modifications from multiple perspectives. To do so, we overcome the unique challenges of GNNs by deriving the changes in the evaluation function from *fundamental principles of calculus*. As a result, our function can measure the influence of both edge deletion and insertion, the latter of which has not been explored in previous studies. To extend the influence function to non-convex GNNs, we establish the proximal Bregman response function [3] for node classification. The influence function derived from this response function relies on weaker assumptions that generally hold for non-convex GNNs. Figure 1 shows an example of influence analysis for a barbell graph from two different perspectives. Through the analysis, one can identify that the edges connecting two different clusters have opposite influences on over-squashing and over-smoothing, providing a unifying view on edge importance.

Experiments on real-world datasets demonstrate that our influence function accurately predicts edge influence in non-convex GNNs. We further show that the influence function is a versatile tool for analyzing various properties of GNNs. We present three practical applications, including 1) an analysis of edge rewiring methods suggested to improve the predictive performance of GNNs, 2) identifying adversarial edge edits that could alter node predictions, and 3) an analysis of edges connecting the nodes with the same label or different labels in terms of the node classification performance.

## 2 Preliminary

### 2.1 Influence function for convex models

Let $\mathcal{L}(x, y, \theta)$ be a loss function, where $x$ is the input, $y$ the label, and $\theta$ the model parameters. The optimal parameters $\theta^*$ minimize the empirical loss over the training set $\mathcal{D}_{\text{train}}$:

$$\theta^* = \arg\min_{\theta} \sum_{(x,y) \in \mathcal{D}_{\text{train}}} \mathcal{L}(x, y, \theta). \tag{1}$$

Influence functions quantify how removing a specific training data point $(x', y')$ impacts model parameters. Computing the exact impact would require retraining the model by excluding the data

point from the training set, which is computationally expensive. Instead, one can model a *response function* that measures the changes in parameters when the data point is upweighted by an amount of $\epsilon \in \mathbb{R}$ by solving the following optimization problem:

$$\theta^*_{x',y',\epsilon} := \arg\min_\theta \frac{1}{N} \sum_{(x,y)\in\mathcal{D}_{\text{train}}} \mathcal{L}(x,y,\theta) + \epsilon\mathcal{L}(x',y',\theta), \tag{2}$$

where $N = |\mathcal{D}_{\text{train}}|$. For example, $\theta^*_{x',y',\epsilon=-1/N}$ corresponds to the optimal parameter obtained from $\mathcal{D}_{\text{train}} \setminus \{x',y'\}$.

The changes in parameter further influence *evaluation function* $f$, such as a validation loss. Koh and Liang [21] demonstrate that when the loss function $\mathcal{L}$ is strictly convex with respect to $\theta$, the derivative of $f\left(\theta^*_{x',y',\epsilon}\right)$ with respect to $\epsilon$ evaluated at $\epsilon = 0$ is:

$$\left.\frac{df\left(\theta^*_{x',y',\epsilon}\right)}{d\epsilon}\right|_{\epsilon=0} = \nabla_\theta f(\theta^*)^\top \left.\frac{d\theta^*_{x',y',\epsilon}}{d\epsilon}\right|_{\epsilon=0} = -\nabla_\theta f(\theta^*)^\top \mathbf{H}_{\theta^*}^{-1} \nabla_\theta\mathcal{L}(x',y',\theta^*), \tag{3}$$

where $\mathbf{H}_{\theta^*} = \frac{1}{N}\sum_{(x,y)\in\mathcal{D}_{\text{train}}} \nabla^2_\theta\mathcal{L}(x,y,\theta^*)$ is the Hessian matrix evaluated at $\theta^*$. Finally, one can approximate the influence of data point $(x',y')$ on the evaluation function $f$ through the linearization around the optimal parameter $\theta^*$ without retraining:

$$f\left(\theta^*_{x',y',-\frac{1}{N}}\right) - f(\theta^*) \approx \frac{1}{N}\nabla_\theta f(\theta^*)^\top \mathbf{H}_{\theta^*}^{-1}\nabla_\theta\mathcal{L}(x',y',\theta^*). \tag{4}$$

When the objective is non-convex, the Hessian can be added with a damping term $\lambda \in \mathbb{R}$, i.e., $\mathbf{H}_{\theta^*} + \lambda I$, leading to a positive definite matrix. We provide a complete derivation in Appendix A.

## 2.2 Influence function for neural networks

Although the influence function in Equation (4) reliably measures the influence of an input for a linear model with a convex objective, the computation is unreliable in practice with deep neural networks, c.f. [4]. Bae et al. [3] identify the three main sources of unreliability related to the standard practices of neural network training and fine-tuning: 1) In non-convex models, the response function is affected more by parameter initialization than by the influence of the data, as gradient methods approximate the solution. 2) The addition of the damping term $\lambda$ works as an $\ell_2$ regularizer of Equation (1) leading to a different response function. 3) The influence function is measured on fully converged parameter $\theta^*$, which is not true in practice due to many reasons, such as early stopping and over-fitting mitigation.

Bae et al. [3] propose a new response function, named the proximal Bregman response function (PBRF), to address the three practices:

$$\theta^*_{\theta_s,x',y',\epsilon} := \arg\min_\theta \frac{1}{N} \sum_{(x,y)\in\mathcal{D}_{\text{train}}} D_\mathcal{L}\left(g_\theta(x), g_{\theta_s}(x), y\right) + \frac{\lambda}{2}\|\theta - \theta_s\|^2 + \epsilon\mathcal{L}(x',y',\theta), \tag{5}$$

where $g_\theta$ is a model parameterized by $\theta$, $\theta_s$ is a reference parameter from which the fine-tuning starts, and $D_\mathcal{L}$ is the Bregman divergence defined as:

$$D_\mathcal{L}\left(h,h',y\right) := \mathcal{L}\left(h,y\right) - \mathcal{L}\left(h',y\right) - \nabla_h\mathcal{L}\left(h',y\right)^\top\left(h - h'\right). \tag{6}$$

For detailed explanations of each term in Equation (5), please refer to Bae et al. [3].

The objective can be further linearized around the model output to simulate the local approximations made in Equation (4), leading to the following influence function[1]:

---

[1]Equation (7) is originally proposed by Teso et al. [31] as an approximation of Equation (4). Bae et al. [3] provide a corresponding response function to the approximation.

$$f\left(\theta_{s,x',y',-\frac{1}{N}}^{*}\right) - f\left(\theta_s\right) \approx \frac{1}{N}\nabla_\theta f(\theta_s)^\top \left(\mathbf{J}_{h\theta_s}^\top \mathbf{H}_{h_s}\mathbf{J}_{h\theta_s} + \lambda\mathbf{I}\right)^{-1}\nabla_\theta \mathcal{L}(x',y',\theta_s), \qquad (7)$$

where $h$ and $h_s$ denote the model outputs parameterized by $\theta$ and $\theta_s$, respectively. $\mathbf{J}_{h\theta_s}$ is the Jacobian of the model outputs with respect to the parameters, and $\mathbf{H}_{h_s}$ is the Hessian of $\mathcal{L}$ with respect to the outputs. When the loss function is convex with respect to the model outputs, the matrix $\mathbf{J}_{h\theta_s}^\top \mathbf{H}_{h_s}\mathbf{J}_{h\theta_s}$ is positive semi-definite. This convexity condition is satisfied by commonly used loss functions, such as cross-entropy and mean squared error. Consequently, for $\lambda > 0$, the matrix $\mathbf{J}_{h\theta_s}^\top \mathbf{H}_{h_s}\mathbf{J}_{h\theta_s} + \lambda\mathbf{I}$ becomes positive definite, ensuring its invertibility. We provide the complete derivation in Appendix A.

## 3 Quantifying the influence of edge edits on GNNs

**Problem setup and notation** We propose an influence function tailored for non-convex GNNs, designed to quantify how edge deletions and insertions perturb model predictions and evaluation functions. We consider node classification on an undirected graph $\mathcal{G} = (\mathcal{V}, \mathcal{E}, \mathbf{X})$, where $\mathcal{V}$ is the set of nodes, $\mathcal{E} \subseteq \mathcal{V} \times \mathcal{V}$ forms the set of edges, and $\mathbf{X} \in \mathbb{R}^{|\mathcal{V}| \times d}$ represents the matrix of node feature $\mathbf{X}_v \in \mathbb{R}^d$ for all node $v \in \mathcal{V}$. We also represent the graph structure using a binary symmetric adjacency matrix $A \in \{0,1\}^{|\mathcal{V}| \times |\mathcal{V}|}$, where $A_{uv} = 1$ indicates an edge between nodes $u$ and $v$. The goal of the node classification is to predict the ground truth label $y_v \in \mathcal{Y}$ of node $v \in \mathcal{V}$. The notation $h_v^{\mathcal{G},\theta}$ refers to the representation of node $v$ obtained by a GNN parameterized by $\theta$ on graph $\mathcal{G}$. A standard supervised training involves a minimization of the average prediction loss $\mathcal{L}(h_v^{\mathcal{G},\theta}, y_v)$ over the entire training set $\mathcal{V}_{\text{train}} \subseteq \mathcal{V}$ with respect to the model parameter $\theta$.

We focus on analyzing the effect of inserting or deleting a single undirected edge $\{u, v\}$. The derivation naturally generalizes to multiple edge edits, which are provided in Appendix C. We first define the reweighted adjacency matrix $A_{uv}^\epsilon$ such that only the $(u, v)$ and $(v, u)$ entries are updated as $A_{uv}^\epsilon = A_{uv} + (2\mathbb{I}[\{u, v\} \in \mathcal{E}] - 1)N\epsilon$, while all other entries remain unchanged. Setting $\epsilon = -1/N$ corresponds to deleting the edge if it exists, or inserting it otherwise. We denote the edge-reweighted graph as $\mathcal{G}^\epsilon = \{\mathcal{V}, \mathcal{E}, A^\epsilon\}$, where we omit the target edge $\{u, v\}$ from the adjacency matrix when it is clear from context for notational simplicity. We collectively refer to edge deletions and insertions as *edge edits*.

**Decomposition of influence function** Let $f(\theta, \mathcal{G})$ be an evaluation function, to which we want to measure the influence of an edge edit. Note that unlike standard parameterization of the evaluation function, i.e., $f(\theta)$, the evaluation function needs to be parameterized by both the model parameter $\theta$ and the graph structure $\mathcal{G}$, because an edge edit not only changes the model parameter but also changes the message propagation paths in GNNs. Due to the dependency of the evaluation function on the graph structure, the derivative of the evaluation function with respect to the change in the weighting of a target edge is decomposed as follows via the chain rule:

$$\frac{df(\theta_\epsilon^*, \mathcal{G}^\epsilon)}{d\epsilon}\bigg|_{\epsilon=0} = \underbrace{\nabla_\theta f(\theta_0^*, \mathcal{G})^\top \frac{\partial \theta_\epsilon^*}{\partial \epsilon}\bigg|_{\epsilon=0}}_{\text{parameter shift}} + \underbrace{\frac{\partial f(\theta, \mathcal{G}^\epsilon)}{\partial A^\epsilon}\frac{\partial A^\epsilon}{\partial \epsilon}\bigg|_{\theta=\theta_0^*,\ \epsilon=0}}_{\text{message propagation}}, \qquad (8)$$

where $\theta_\epsilon^*$ represents the response function of an edge edit; a formal definition is provided in the following paragraph.

**Remark.** *The influence functions proposed for GNNs in Chen et al. [7], Wu et al. [35] only consider the effect of parameter shift while missing the changes in the message propagation path.*

**Parameter shift** To quantify the change in model parameters for non-convex GNNs, we propose a graph-adapted version of the PBRF. The original PBRF, introduced in Equation (5), only quantifies the changes in parameters when the weight of a single data point is modified, inapplicable to our scenario where the weight of an edge changes. To address this challenge, we introduce an edge-edit PBRF that explicitly accounts for changes in node representations caused by an edge edit.

We define the edge-edit PBRF as follows[2]:

$$\theta_\epsilon^* := \arg\min_\theta \frac{1}{N} \sum_{v \in \mathcal{V}_{\text{train}}} D_{\mathcal{L}} \left( h_v^{\mathcal{G},\theta}, h_v^{\mathcal{G},\theta_s} \right) + \frac{\lambda}{2} \|\theta - \theta_s\|^2 + \sum_{v \in \mathcal{V}_{\text{train}}} \epsilon \left( \mathcal{L} \left( h_v^{\mathcal{G},\theta} \right) - \mathcal{L} \left( h_v^{\mathcal{G}^{-\frac{1}{N}},\theta} \right) \right). \tag{9}$$

The first two terms regularize $\theta$ to stay close to the reference parameter $\theta_s$, both in terms of the output space and the parameter space. The final term, with $\epsilon < 0$, encourages $\theta$ to increase the loss on the original graph while decreasing the loss on the edge-edited graph. Thus, edge-edit PBRF can be interpreted as identifying parameters near $\theta_s$ that fail to predict correctly on the original graph but succeed on the edge-edited graph, thereby responding to the edge edit. The scalar $\epsilon$ controls the magnitude of this response to the edge edit. Note that $\theta_0^* = \theta_s$, since the Bregman divergence $D_{\mathcal{L}}(h, h', y)$ is minimized when $h = h'$.

Based on the edge-edit PBRF objective, the changes in the evaluation function caused by the parameter shift are then given by:

$$\nabla_\theta f(\theta_0^*, \mathcal{G})^\top \left. \frac{\partial \theta_\epsilon^*}{\partial \epsilon} \right|_{\epsilon=0} = -\nabla_\theta f(\theta_s, \mathcal{G})^\top \mathbf{G}^{-1} \sum_{v \in \mathcal{V}_{\text{train}}} \left( \nabla_\theta \mathcal{L} \left( h_v^{\mathcal{G},\theta_s} \right) - \nabla_\theta \mathcal{L} \left( h_v^{\mathcal{G}^{-\frac{1}{N}},\theta_s} \right) \right). \tag{10}$$

where $\mathbf{G} = \mathbf{J}_{h\theta_s}^\top \mathbf{H}_{h_s} \mathbf{J}_{h\theta_s} + \lambda \mathbf{I}$ denotes the generalized Gauss–Newton Hessian with a damping term, and $h$ and $\tilde{h}_s$ denote the node representations obtained using parameters $\theta$ and $\theta_s$, respectively. A detailed derivation is provided in Appendix B.

**Message propagation**   To quantify the changes in the evaluation function caused by the modification in the message propagation path, we can further expand the message propagation term in Equation (8). Since $\partial A_{ij}^\epsilon / \partial \epsilon = 0$ for all $\{i, j\} \neq \{u, v\}$, the message propagation term simplifies as follows:

$$\frac{\partial f(\theta, \mathcal{G}^\epsilon)}{\partial A^\epsilon} \left. \frac{\partial A^\epsilon}{\partial \epsilon} \right|_{\theta=\theta_0^*,\, \epsilon=0} = \left. \frac{\partial f(\theta, \mathcal{G}^\epsilon)}{\partial A_{uv}^\epsilon} \frac{\partial A_{uv}^\epsilon}{\partial \epsilon} \right|_{\theta=\theta_s,\, \epsilon=0} + \left. \frac{\partial f(\theta, \mathcal{G}^\epsilon)}{\partial A_{vu}^\epsilon} \frac{\partial A_{vu}^\epsilon}{\partial \epsilon} \right|_{\theta=\theta_s,\, \epsilon=0}$$

$$= (2\mathbb{I}[\{u, v\} \in \mathcal{E}] - 1) N \left( \frac{\partial f(\theta_s, \mathcal{G})}{\partial A_{uv}} + \frac{\partial f(\theta_s, \mathcal{G})}{\partial A_{vu}} \right). \tag{11}$$

**Unified influence function under edge edits**   By substituting Equation (11) and Equation (10) into Equation (8) and linearizing around $\epsilon = 0$, we obtain a first-order approximation of the influence function that captures both parameter shift and message propagation effects. The resulting influence of an edge edit is given by:

$$f \left( \theta_{-\frac{1}{N}}^*, \mathcal{G}^{-\frac{1}{N}} \right) - f(\theta_s, \mathcal{G}) \approx \frac{1}{N} \nabla_\theta f(\theta_s, \mathcal{G})^\top \mathbf{G}^{-1} \sum_{v \in \mathcal{V}_{\text{train}}} \left( \nabla_\theta \mathcal{L} \left( h_v^{\mathcal{G},\theta_s} \right) - \nabla_\theta \mathcal{L} \left( h_v^{\mathcal{G}^{-\frac{1}{N}},\theta_s} \right) \right)$$

$$- (2\mathbb{I}[\{u, v\} \in \mathcal{E}] - 1) \left( \frac{\partial f(\theta_s, \mathcal{G})}{\partial A_{uv}} + \frac{\partial f(\theta_s, \mathcal{G})}{\partial A_{vu}} \right). \tag{12}$$

Directly computing $\mathbf{G}^{-1}$ is computationally infeasible for large models. To address this, we approximate the inverse Hessian-vector product $\mathbf{G}^{-1} \nabla_\theta f(\theta_s, \mathcal{G})$ using the LiSSA algorithm [1], a stochastic iterative method. A detailed description is provided in Appendix D.

## 4   Validation of influence function

We measure the correctness of the proposed influence function on three different evaluation metrics: over-squashing and over-smoothing measures, and a validation loss. Our goal is to precisely predict

---

[2]For notational simplicity, we omit the label $y_v$ of node $v$ in expressions involving the loss function or the Bregman divergence, as it is clear from context.

how much over-squashing and over-smoothing measures, and validation loss change when an existing edge is deleted from the graph or when a potential edge is added between two nodes.

## 4.1 Evaluation functions

We explain the three evaluation functions considered in detail.

**Over-squashing**    Over-squashing [2] is the phenomenon in which information from distant nodes is overly compressed during message passing, preventing it from effectively influencing node representations. Topping et al. [32] propose a gradient-based metric $\partial h_v / \partial \mathbf{X}_u$ to quantify the influence of the initial node feature $\mathbf{X}_u$ on the node representation $h_v$. One way to measure over-squashing in a graph is by averaging the gradients between all pairs of distant nodes. However, computing these gradients for every such pair is computationally expensive. Moreover, estimating the influence function requires taking the derivative of this measurement, which is even more costly due to the complexity of the measurement itself. To address this issue, we propose an alternative over-squashing measure similar to the gradient-based one but without derivation. Let $\mathcal{N}_L(v)$ be the set of nodes that can be reached from node $v$ in exactly $L$ hops. To measure the influence of node $u$ to node $v$ in an $L$-layer GNN, we first define modified graph $\mathcal{G}'(v) = \{\mathcal{V}, \mathcal{E}, \mathbf{X}'\}$, where

$$\mathbf{X}'_u = \begin{cases} \mathbf{0}, & \text{if } u \in \mathcal{N}_L(v), \\ \mathbf{X}_u, & \text{otherwise,} \end{cases} \tag{13}$$

for all $u$ in $\mathcal{V}$. With the modified graph, we propose a new over-squashing measure as:

$$f_{\text{OQ}}(\theta, \mathcal{G}) = \sum_{v \in \mathcal{V}} \left\| h_v^{\mathcal{G},\theta} - h_v^{\mathcal{G}'(v),\theta} \right\|_2. \tag{14}$$

The measure computes the average difference in node representations with and without $L$-hop neighborhood node features. It is similar to the gradient-based metric in that both quantify how a node's representation changes when input features of other nodes are modified, although one does so through gradient computation and the other through direct input masking.

**Over-smoothing**    We use the Dirichlet energy that quantifies the over-smoothing phenomenon in GNNs [23]. Dirichlet energy is defined as the average squared $\ell_2$ distance between the embeddings of adjacent nodes [6], and serves as a proxy for the representational diversity across the graph. A lower Dirichlet energy indicates more severe over-smoothing, as node representations become increasingly indistinguishable due to excessive message passing. Conversely, higher Dirichlet energy suggests that node embeddings remain more discriminative.

**Validation loss**    We use a standard mean cross-entropy loss on the validation set as an evaluation function.

## 4.2 Actual vs. predicted influence

**Datasets and experimental setup**    We conduct experiments on five datasets: the citation graphs Cora, Citeseer, and Pubmed [29, 36], where the task is to predict each paper's research area based on citation relationships; and the Wikipedia graphs Chameleon and Squirrel [28], where the task is to estimate page traffic based on hyperlink relationships. For the citation graphs, we follow the data splits provided by Yang et al. [36], and for the Wikipedia graphs, we use the splits from Pei et al. [27].

We evaluate the prediction performance of our method on three representative graph neural networks: GCN [20], GAT [33], and ChebNet [13]. We compare our approach with GIF, an existing graph influence function [7, 35]. To measure the actual influence, we first train the model on the original graph. We then retrain the GNN on the edge-edited graph by optimizing the minimization objective defined by each response function. Specifically, for our influence function, we fine-tuned the model by minimizing the objective in Equation (9), using the original model parameters as $\theta_s$ and setting $\epsilon = -1/N$. For GIF, we retrain the model by minimizing the loss on the edge-edited graph, using

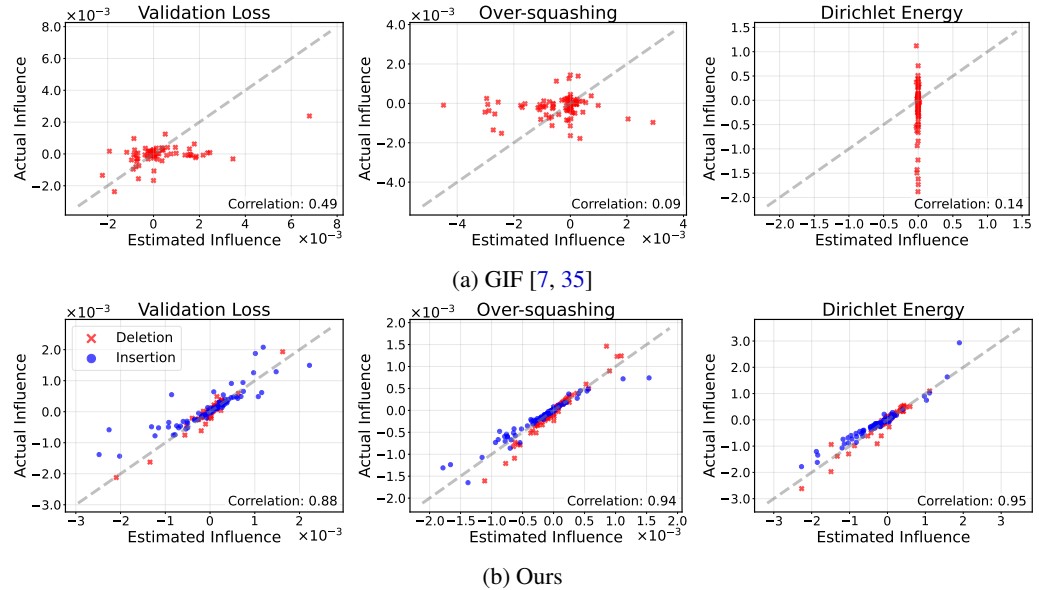

(a) GIF [7, 35]

(b) Ours

Figure 2: Predicted influence versus actual influence on a four-layer GCN. The x-axis represents the predicted influence, the y-axis represents the actual influence, and the dotted line represents the perfect alignment.

the same parameter initialization as the original model. The actual influence is computed as the difference between the evaluation values of the original model and the retrained model.

**Results** Figure 2 presents scatter plots for the Cora dataset, where each point shows the actual and predicted influence of a single edge edit. The x-axis represents the predicted influence computed by each method, and the y-axis represents the actual influence. The first row illustrates results from GIF [7, 35], while the second row shows results from our proposed method for edge deletions and insertions, respectively. Additional results for other GNNs and datasets are provided in Appendix E.

GIF [7, 35] fails to reliably estimate actual influences on non-convex GNNs, achieving correlations of only $0.09$ and $0.14$ for over-squashing and Dirichlet energy, respectively. In contrast, our proposed method significantly improves prediction, achieving correlations up to $0.95$, closely aligning with the ideal predictions shown by the grey dotted line. Moreover, our method exhibits strong predictive capability for both edge deletions and insertions.

We further evaluate how editing the graph to improve each measurement affects prediction on the test nodes. To perform the edge edits, we first compute the influence function with respect to each measurement and then edit the edges that are predicted to improve the corresponding measurement. For validation loss, we edit the edges with the $k$ smallest influence values, as negative influence indicates that removing the edge is expected to reduce the measurement. For the other measurements, we edit the edges with the $k$ largest influence values. The value of $k$ is selected to maximize validation accuracy. We compare the edge edits produced by our influence function to two baselines:

Table 1: Test accuracy on edge-edited graphs. The best result is highlighted in bold.

|  | Cora | CiteSeer | PubMed |
|---|---|---|---|
| GCN | 81.0±0.3 | 69.3±0.5 | 75.6±1.0 |
| Random | 81.1±0.4 | 69.2±0.4 | 75.7±0.8 |
| GIF | 80.9±0.5 | 69.2±0.5 | 75.6±0.9 |
| Ours (DE) | 80.8±0.4 | 69.5±0.5 | 75.4±1.2 |
| Ours ($f_{OQ}$) | 81.1±0.4 | 69.3±0.5 | 75.4±1.0 |
| Ours (VL) | **82.1±0.5** | **69.6±0.7** | **76.4±1.3** |

random edge deletion (Random) and edge edits using GIF [7, 35] in place of our influence function.

Table 1 shows the test accuracy when the model is trained and evaluated on edge-edited graphs. Dirichlet energy (DE), over-squashing ($f_{OQ}$), and validation loss (VL) are used as shorthand in the table. Our edits based on validation loss achieve the best performance across three datasets. This result suggests that editing edges to reduce validation loss can be helpful for improving test accuracy. In contrast, editing edges to improve Dirichlet energy or the over-squashing measurement does not

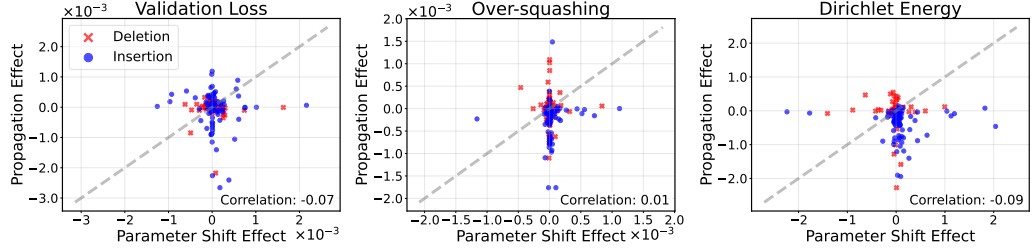

Figure 3: The relationship between parameter shift effect and message propagation effect defined in Equation (8). The x-axis denotes the parameter shift effect, and the y-axis denotes the message propagation effect.

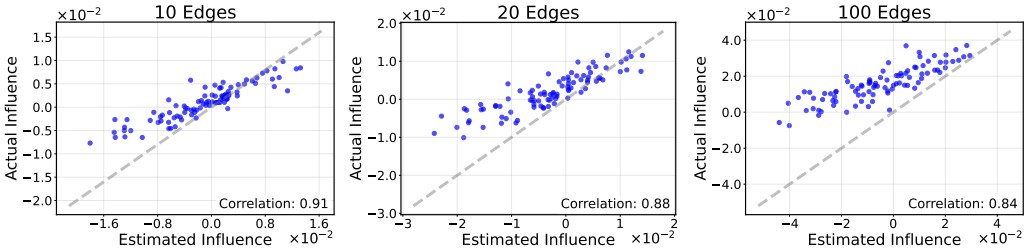

Figure 4: Predicted influence versus actual influence on a four-layer GCN under varying numbers of inserted edges.

consistently improve test accuracy. This indicates that optimizing these intermediate metrics does not necessarily translate to better predictive performance. Finally, editing edges to reduce validation loss using GIF fails to improve test accuracy, which we attribute to its inaccurate influence estimation on non-convex GNNs.

**Analysis on the influence of message propagation** We analyze the importance of explicitly incorporating the influence of message propagation when predicting the influence of edge edits. If the influence of the message propagation is negligible or highly correlated with the influence of the parameter shift, predicting the influence of the message propagation may not be necessary. To validate the importance of the message propagation influence, we measure its correlation with the influence of parameter shift.

Figure 3 shows scatter plots comparing these two influences on the Cora dataset. We observe a low correlation between the message propagation and parameter shift influence across all metrics, while their magnitudes remain comparable. These results underscore the necessity of explicitly measuring the influence of message propagation in estimating the influence of edge edits.

**Influence estimation for multiple edge edits** We evaluate the accuracy of the proposed influence function in predicting the actual influence under multiple edge edits. Figure 4 presents scatter plots of predicted versus actual influence when inserting different numbers of edges. The experiments are conducted on the Cora dataset, with validation loss used as the evaluation metric. Specifically, we report results for simultaneous insertions of 10, 20, and 100 edges.

Our influence function maintains a high correlation with the actual influence, achieving 0.84 even under 100 simultaneous edge edits. Nonetheless, Figure 4 shows that the accuracy of influence estimation diminishes as the number of simultaneous edits increases. We attribute this degradation to first-order approximation errors induced by substantial parameter shifts, a phenomenon that has also been consistently observed in prior studies on group influence estimation [22, 5, 30].

## 5 Applications

**Influence function as a tool for adversarial attack** Once we identify the most influential edges in the entire input graph, we can significantly affect the model's output by editing them. This process

Table 2: Test accuracy under adversarial edge edits. The attack ratio denotes the percentage of edges added or removed with respect to the total number of edges in the original graph. The values in parentheses next to each dataset name indicate the test accuracy on the original graph before any edits.

| Attack ratio | Cora (81.0) | | | CiteSeer (69.3) | | | PubMed (75.6) | | |
|---|---|---|---|---|---|---|---|---|---|
| | 1% | 3% | 5% | 1% | 3% | 5% | 1% | 3% | 5% |
| DICE | 80.8±0.4 | 80.3±0.5 | 80.0±0.5 | 69.0±0.7 | 68.5±0.5 | 68.1±0.5 | 75.1±0.9 | 74.5±1.4 | 73.5±1.1 |
| PRBCD | 80.6±0.5 | 79.4±0.5 | 78.7±0.9 | 68.5±0.5 | 67.4±0.8 | 66.4±0.5 | 74.6±1.3 | 72.9±1.8 | 70.9±1.7 |
| GIF | 80.8±0.4 | 80.8±0.6 | 80.6±0.4 | 69.8±0.5 | 69.8±0.7 | 69.3±0.5 | 75.4±1.1 | 75.3±1.0 | 75.2±0.9 |
| Ours (DE) | 81.0±0.3 | 80.7±0.3 | 80.8±0.3 | 69.1±0.5 | 69.2±0.7 | 68.8±0.8 | 75.6±1.0 | 75.2±1.0 | 75.0±1.2 |
| Ours ($f_{OQ}$) | 80.9±0.4 | 80.9±0.6 | 80.6±0.7 | 68.9±0.6 | 69.0±0.5 | 68.6±0.8 | 75.4±0.9 | 74.9±1.0 | 74.4±1.1 |
| Ours (VL) | **80.2±0.7** | **79.1±0.7** | **78.4±0.8** | **68.2±0.5** | **66.7±0.6** | **65.1±0.9** | **73.2±1.4** | **71.2±1.5** | **69.9±1.1** |

represents a form of global adversarial attack for the graph, aiming to reduce the performance of the entire graph.

Table 2 presents the adversarial attack performance of our methods, comparing it to previous influence-function-based methods and other adversarial attack methods, DICE [34] and PRBCD [14], in the white-box scenario. As a result, the GIF shows poor performance, while our method with a validation loss metric outperforms the others. An interesting observation is that, among the three metrics we tested, the validation loss-based attack is more effective than the tailored attack methods.

**Explanation of the characteristics of beneficial edges** We demonstrate the influence of edges connecting the node with the same label (*homophilic edge*) and different labels (*heterophilic edge*) on GCN. Figure 5 presents a dumbbell plot showing the mean influence of homophilic and heterophilic edges across six datasets, which include three homophilic graphs (Cora, CiteSeer, and PubMed), where edges tend to connect nodes with the same labels, and three heterophilic graphs (Chameleon, Actor, and Squirrel), where edges tend to connect nodes with different labels.

For both types of graphs, adding homophilic edges is more beneficial than adding heterophilic ones. The opposite effect is observed for edge deletion. These findings suggest that increasing a graph's homophily is beneficial, which aligns with the homophilic nature of the tested GNNs.

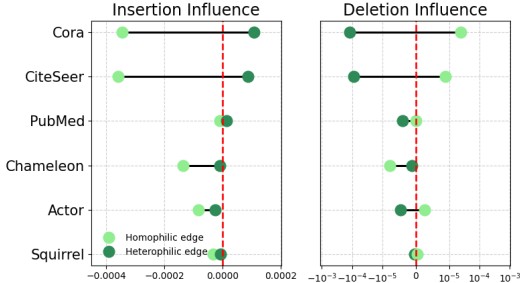

Figure 5: Mean influence of homophilic and heterophilic edges on validation loss for edge insertion (left) and edge deletion (right). Each dumbbell connects the average influence of homophilic (light green) and heterophilic (dark green) edges across six datasets. A negative value indicates that the edge edit decreases the validation loss, thus improving the performance.

**Analysis of the effect of edge rewiring** Our approach provides multiple analytical perspectives on existing rewiring strategies through various evaluation metrics. Specifically, we analyze BORF [26] and FoSR [19]. BORF alleviates over-squashing by inserting edges between nodes with negative curvature and mitigates over-smoothing by removing edges with positive curvature, while FoSR alleviates over-squashing by inserting edges that enlarge the spectral gap.

Figure 6 presents the estimated influence of edges selected by BORF and FoSR from three measurement perspectives. Edge insertions chosen by both methods generally increase the over-squashing measurement, while edge deletions in BORF tend to increase the over-smoothing measurement, confirming that each method effectively targets its intended GNN challenge. Nevertheless, we also observe unintended side effects: edge insertions often exacerbate over-smoothing, and neither insertions nor deletions consistently reduce validation loss. Mitigating these side effects could substantially enhance the overall effectiveness and reliability of edge rewiring methods.

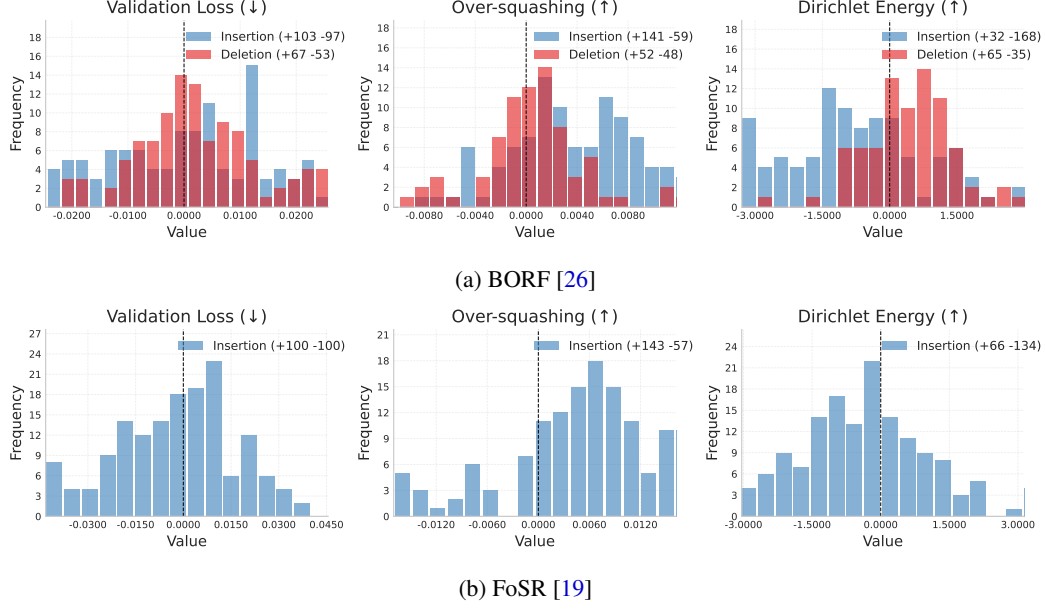

Figure 6: Histograms of the estimated influence of edge insertions (blue) and deletions (red) selected by BORF [26] and FoSR [19], evaluated on a four-layer GCN trained on the Texas dataset. Influences are measured across validation loss, over-squashing, and over-smoothing. Arrows ($\downarrow$ / $\uparrow$) indicate the desired direction of each measurement (decrease/increase).

# 6 Related work

Originally introduced by Cook and Weisberg [12], the influence function quantifies how removing a single training point would affect model parameters after retraining. Koh and Liang [21] adapted this concept to modern machine learning, providing efficient approximations of the influence on model predictions without the need for retraining. Influence functions have since been widely employed for model interpretability [17, 16], data valuation [8, 18], adversarial analysis [11, 10], and unlearning [39, 37].

A significant limitation of influence functions is the assumption of strict convexity in the loss function [4], which does not hold for many deep learning models. To overcome this, Teso et al. [31] approximate the Hessian using the Fisher information matrix, while Bae et al. [3] introduce a proximal Bregman response function objective that relaxes this convexity requirement. Recent studies have further extended influence functions to diffusion models [25] and large language models [15, 9, 38]. In graph settings, Chen et al. [7] and Wu et al. [35] apply influence functions to transductive node classification, and Song et al. [30] analyze group-level influence.

# 7 Conclusion

We introduce an enhanced graph influence function that estimates the impact of edge perturbations on model predictions in Graph Neural Networks. Unlike existing graph influence functions, our approach explicitly incorporates message propagation effects and relaxes the convexity assumption, enabling it to be applied to commonly used non-convex GNNs. Additionally, we extend the framework to handle both edge deletions and insertions, broadening its applicability to real-world graph rewiring tasks. Experimental results on various real-world datasets demonstrate that our method provides significantly more accurate estimates of influence than previous methods and enables effective improvements in key target measurements such as validation loss, over-squashing, and over-smoothing. Despite these advances, our method has limitations. Although it achieves high accuracy under multiple edge edits, performance gradually declines as the number of simultaneous edits grows, and scalability to deep GNNs remains challenging. Future work could address these issues to enhance the applicability of the method to complex graph learning tasks.

## Acknowledgements

This work was supported by the National Research Foundation of Korea (NRF) grant funded by the Korea government(MSIT) (RS-2024-00337955; RS-2023-00217286) and Institute of Information & communications Technology Planning & Evaluation (IITP) grant funded by the Korea government(MSIT) (RS-2024-00457882, National AI Research Lab Project; RS-2019-II191906, Artificial Intelligence Graduate School Program(POSTECH)).

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

# A Derivation of the influence function

In this section, we derive the influence function for linear models [21], as shown in Equation (3), and for neural networks [3], as shown in Equation (7). In particular, we reformulate the derivation by Bae et al. [3] to fit our setting.

## A.1 Influence function for linear models

We begin by considering the re-weighted objective introduced in Equation (2):

$$\mathcal{J}(\theta, \epsilon) = \frac{1}{N} \sum_{(x,y) \in \mathcal{D}_{\text{train}}} \mathcal{L}(x, y, \theta) + \epsilon \mathcal{L}(x', y', \theta). \tag{15}$$

Assuming that the loss function $\mathcal{L}$ is twice continuously differentiable and that $\mathcal{J}(\theta, \epsilon)$ is strictly convex in $\theta$, the optimal parameter $\theta^*_{x',y',\epsilon}$ minimizing the objective satisfies the first-order optimality condition:

$$\nabla_\theta \mathcal{J}(\theta^*_{x',y',\epsilon}, \epsilon) = 0. \tag{16}$$

To justify that the response function $\theta^*_{x',y',\epsilon}$ is differentiable with respect to $\epsilon$, we apply the Implicit Function Theorem to the optimality condition. The following conditions must be satisfied for the theorem to apply:

- The function $\nabla_\theta \mathcal{J}(\theta, \epsilon)$ is continuously differentiable in both $\theta$ and $\epsilon$. This holds because $\mathcal{J}(\theta, \epsilon)$ is constructed as a linear combination of smooth loss functions, and its dependence on $\epsilon$ is linear.

- The optimality condition $\nabla_\theta \mathcal{J}(\theta^*_{x',y',\epsilon}, \epsilon) = 0$ holds by definition, since $\theta^*_{x',y',\epsilon}$ minimizes the objective $\mathcal{J}(\theta, \epsilon)$.

- The Hessian $\nabla^2_\theta \mathcal{J}(\theta, \epsilon)$, taken with respect to $\theta$, is non-singular in a neighborhood of $\epsilon = 0$, as $\mathcal{J}(\theta, \epsilon)$ is assumed to be strictly convex in $\theta$.

Under these conditions, the Implicit Function Theorem ensures that $\theta^*_{x',y',\epsilon}$ is continuously differentiable with respect to $\epsilon$, and we differentiate Equation (16) using the chain rule:

$$\frac{d}{d\epsilon} \left( \nabla_\theta \mathcal{J}(\theta^*_{x',y',\epsilon}, \epsilon) \right) = \nabla^2_\theta \mathcal{J}(\theta^*_{x',y',\epsilon}, \epsilon) \cdot \frac{d\theta^*_{x',y',\epsilon}}{d\epsilon} + \nabla_\theta \mathcal{L}(x', y', \theta^*_{x',y',\epsilon}) = 0. \tag{17}$$

Solving for the derivative yields:

$$\frac{d\theta^*_{x',y',\epsilon}}{d\epsilon} = - \left( \nabla^2_\theta \mathcal{J}(\theta^*_{x',y',\epsilon}, \epsilon) \right)^{-1} \nabla_\theta \mathcal{L}(x', y', \theta^*_{x',y',\epsilon}). \tag{18}$$

Evaluating at $\epsilon = 0$, we obtain the influence function:

$$\left. \frac{d\theta^*_{x',y',\epsilon}}{d\epsilon} \right|_{\epsilon=0} = - \left( \nabla^2_\theta \mathcal{J}(\theta^*, 0) \right)^{-1} \nabla_\theta \mathcal{L}(x', y', \theta^*), \tag{19}$$

where $\theta^* := \theta^*_{x',y',\epsilon=0}$ is the minimizer of the original objective without perturbation.

## A.2 Influence function for neural networks

Bae et al. [3] demonstrated that the influence function using the generalized Gauss-Newton Hessian corresponds to that of the linearized form of the proximal Bregman response function objective[3]:

$$\theta^*_{\text{lin},x',y',\epsilon} := \arg\min_{\theta} \frac{1}{N} \sum_{(x,y)\in\mathcal{D}_{\text{train}}} D_{\mathcal{L}_{\text{quad}}}\left(h_x^{\text{lin},\theta}, h_x^{\theta_s}\right) + \frac{\lambda}{2} \|\theta - \theta_s\|^2 + \epsilon \nabla_\theta \mathcal{L}(x',\theta_s)^\top \theta, \quad (20)$$

where $h_x^\theta = g_\theta(x)$, and $\mathcal{L}_{\text{quad}}$ and $h_x^{\text{lin},\theta}$ are the quadratic and linear approximations of the loss and model output, respectively:

$$\mathcal{L}_{\text{quad}}(h_x^\theta) = \mathcal{L}(h_x^{\theta_s}) + \nabla_{h_x^\theta} \mathcal{L}(h_x^{\theta_s})^\top (h_x^\theta - h_x^{\theta_s}) + \frac{1}{2}(h_x^\theta - h_x^{\theta_s})^\top \nabla^2_{h_x^\theta} \mathcal{L}(h_x^{\theta_s})(h_x^\theta - h_x^{\theta_s}),$$

$$h_x^{\text{lin},\theta} = h_x^{\theta_s} + \mathbf{J}_{h_x^\theta \theta_s}(\theta - \theta_s), \quad (21)$$

where $\mathbf{J}_{h_x^\theta \theta_s} := \frac{\partial h_x^\theta}{\partial \theta}\big|_{\theta=\theta_s}$ is the Jacobian of the model output with respect to the parameters.

We now expand the Bregman divergence term. First, using $\mathcal{L}_{\text{quad}}(h_x^{\theta_s}) = \mathcal{L}(h_x^{\theta_s})$:

$$D_{\mathcal{L}_{\text{quad}}}(h_x^{\text{lin},\theta}, h_x^{\theta_s}) = \mathcal{L}_{\text{quad}}(h_x^{\text{lin},\theta}) - \mathcal{L}_{\text{quad}}(h_x^{\theta_s}) - \nabla_{h_x^\theta} \mathcal{L}_{\text{quad}}(h_x^{\theta_s})^\top (h_x^{\text{lin},\theta} - h_x^{\theta_s})$$

$$= \mathcal{L}(h_x^{\theta_s}) + \nabla_{h_x^\theta} \mathcal{L}(h_x^{\theta_s})^\top (h_x^\theta - h_x^{\theta_s}) + \frac{1}{2}(h_x^\theta - h_x^{\theta_s})^\top \nabla^2_{h_x^\theta} \mathcal{L}(h_x^{\theta_s})(h_x^\theta - h_x^{\theta_s})$$

$$- \mathcal{L}(h_x^{\theta_s}) - \nabla_{h_x^\theta} \mathcal{L}(h_x^{\theta_s})^\top (h_x^\theta - h_x^{\theta_s}). \quad (22)$$

Next, using $h_x^{\text{lin},\theta} - h_x^{\theta_s} = \mathbf{J}_{h_x^\theta \theta_s}(\theta - \theta_s)$:

$$D_{\mathcal{L}_{\text{quad}}}(h_x^{\text{lin},\theta}, h_x^{\theta_s}) = \nabla_{h_x^\theta} \mathcal{L}(h_x^{\theta_s})^\top \mathbf{J}_{h_x^\theta \theta_s}(\theta - \theta_s) + \frac{1}{2}(\theta - \theta_s)^\top \mathbf{J}_{h_x^\theta \theta_s}^\top \nabla^2_{h_x^\theta} \mathcal{L}(h_x^{\theta_s}) \mathbf{J}_{h_x^\theta \theta_s}(\theta - \theta_s)$$

$$- \nabla_{h_x^\theta} \mathcal{L}(h_x^{\theta_s})^\top \mathbf{J}_{h_x^\theta \theta_s}(\theta - \theta_s)$$

$$= \frac{1}{2}(\theta - \theta_s)^\top \mathbf{J}_{h_x^\theta \theta_s}^\top \nabla^2_{h_x^\theta} \mathcal{L}(h_x^{\theta_s}) \mathbf{J}_{h_x^\theta \theta_s}(\theta - \theta_s). \quad (23)$$

Since $\theta^*_{\text{lin},x',y',\epsilon}$ is the optimal solution, the gradient of the objective with respect to $\theta$ is zero at this point:

$$0 = -\frac{1}{N} \sum_{(x,y)\in\mathcal{D}_{\text{train}}} \mathbf{J}_{h_x^\theta \theta_s}^\top \nabla^2_{h_x^\theta} \mathcal{L}(h_x^{\theta_s}) \mathbf{J}_{h_x^\theta \theta_s}(\theta^*_{\text{lin},x',y',\epsilon} - \theta_s) + \lambda(\theta^*_{\text{lin},x',y',\epsilon} - \theta_s) + \epsilon \nabla_\theta \mathcal{L}(x',\theta_s).$$

$$(24)$$

Solving for $\theta^*_{\text{lin},x',y',\epsilon}$, we obtain:

$$\theta^*_{\text{lin},x',y',\epsilon} = \theta_s + \left(\mathbf{J}_{h^\theta \theta_s}^\top \mathbf{H}_{h_s} \mathbf{J}_{h^\theta \theta_s} + \lambda \mathbf{I}\right)^{-1} \nabla_\theta \mathcal{L}(x',\theta_s)\epsilon, \quad (25)$$

where $\mathbf{J}_{h^\theta \theta_s}^\top \mathbf{H}_{h_s} \mathbf{J}_{h^\theta \theta_s} := \frac{1}{N} \sum_{(x,y)\in\mathcal{D}_{\text{train}}} \mathbf{J}_{h_x^\theta \theta_s}^\top \nabla^2_{h_x^\theta} \mathcal{L}(h_x^{\theta_s}) \mathbf{J}_{h_x^\theta \theta_s}$ is the generalized Gauss-Newton Hessian.

---

[3]For notational simplicity, we omit the label $y$ in expressions involving the loss function or the Bregman divergence, as it is clear from context.

# B   Gradient derivation for edge-edit PBRF

Similar to the derivation of the influence function for the PBRF objective in Appendix A.2, we demonstrate that the influence function in Equation (10) corresponds to that of the linearized form of the edge-edit PBRF objective:

$$\theta^*_{\text{lin},\epsilon} := \arg\min_\theta \frac{1}{N} \sum_{v \in \mathcal{V}_{\text{train}}} D_{\mathcal{L}_{\text{quad}}}\left(h_v^{\mathcal{G},\theta,\text{lin}}, h_v^{\mathcal{G},\theta_s}\right) + \frac{\lambda}{2} \|\theta - \theta_s\|^2$$
$$+ \sum_{v \in \mathcal{V}_{\text{train}}} \epsilon \left(\nabla_\theta \mathcal{L}(h_v^{\mathcal{G},\theta_s}) - \nabla_\theta \mathcal{L}(h_v^{\mathcal{G}-\frac{1}{N},\theta_s})\right)^\top \theta, \tag{26}$$

where $\mathcal{L}_{\text{quad}}$ and $h_v^{\mathcal{G},\theta,\text{lin}}$ denote the quadratic and linear approximations of the loss and the model output, respectively:

$$\mathcal{L}_{\text{quad}}(h_v^{\mathcal{G},\theta}) = \mathcal{L}(h_v^{\mathcal{G},\theta_s}) + \nabla_{h_v^{\mathcal{G},\theta}} \mathcal{L}(h_v^{\mathcal{G},\theta_s})^\top (h_v^{\mathcal{G},\theta} - h_v^{\mathcal{G},\theta_s})$$
$$+ \frac{1}{2}(h_v^{\mathcal{G},\theta} - h_v^{\mathcal{G},\theta_s})^\top \nabla^2_{h_v^{\mathcal{G},\theta}} \mathcal{L}(h_v^{\mathcal{G},\theta_s})(h_v^{\mathcal{G},\theta} - h_v^{\mathcal{G},\theta_s}),$$
$$h_v^{\mathcal{G},\theta,\text{lin}} = h_v^{\mathcal{G},\theta_s} + \mathbf{J}_{h_v^{\mathcal{G},\theta}\theta_s}(\theta - \theta_s), \tag{27}$$

where $\mathbf{J}_{h_v^{\mathcal{G},\theta}\theta_s} := \frac{\partial h_v^{\mathcal{G},\theta}}{\partial \theta}\big|_{\theta=\theta_s}$ is the Jacobian of the model output with respect to the parameters.

We first expand the Bregman divergence term:

$$D_{\mathcal{L}_{\text{quad}}}(h_v^{\mathcal{G},\theta,\text{lin}}, h_v^{\mathcal{G},\theta_s}) = \mathcal{L}_{\text{quad}}(h_v^{\mathcal{G},\theta,\text{lin}}) - \mathcal{L}_{\text{quad}}(h_v^{\mathcal{G},\theta_s}) - \nabla_{h_v^{\mathcal{G},\theta}} \mathcal{L}_{\text{quad}}(h_v^{\mathcal{G},\theta_s})^\top (h_v^{\mathcal{G},\theta,\text{lin}} - h_v^{\mathcal{G},\theta_s})$$
$$= \mathcal{L}(h_v^{\mathcal{G},\theta_s}) + \nabla_{h_v^{\mathcal{G},\theta}} \mathcal{L}(h_v^{\mathcal{G},\theta_s})^\top (h_v^{\mathcal{G},\theta} - h_v^{\mathcal{G},\theta_s})$$
$$+ \frac{1}{2}(h_v^{\mathcal{G},\theta} - h_v^{\mathcal{G},\theta_s})^\top \nabla^2_{h_v^{\mathcal{G},\theta}} \mathcal{L}(h_v^{\mathcal{G},\theta_s})(h_v^{\mathcal{G},\theta} - h_v^{\mathcal{G},\theta_s})$$
$$- \mathcal{L}(h_v^{\mathcal{G},\theta_s}) - \nabla_{h_v^{\mathcal{G},\theta}} \mathcal{L}(h_v^{\mathcal{G},\theta_s})^\top (h_v^{\mathcal{G},\theta} - h_v^{\mathcal{G},\theta_s}). \tag{28}$$

Using $h_v^{\mathcal{G},\theta,\text{lin}} - h_v^{\mathcal{G},\theta_s} = \mathbf{J}_{h_v^{\mathcal{G},\theta}\theta_s}(\theta - \theta_s)$, we substitute into the expression:

$$D_{\mathcal{L}_{\text{quad}}}(h_v^{\mathcal{G},\theta,\text{lin}}, h_v^{\mathcal{G},\theta_s}) = \nabla_{h_v^{\mathcal{G},\theta}} \mathcal{L}(h_v^{\mathcal{G},\theta_s})^\top \mathbf{J}_{h_v^{\mathcal{G},\theta}\theta_s}(\theta - \theta_s)$$
$$+ \frac{1}{2}(\theta - \theta_s)^\top \mathbf{J}^\top_{h_v^{\mathcal{G},\theta}\theta_s} \nabla^2_{h_v^{\mathcal{G},\theta}} \mathcal{L}(h_v^{\mathcal{G},\theta_s}) \mathbf{J}_{h_v^{\mathcal{G},\theta}\theta_s}(\theta - \theta_s)$$
$$- \nabla_{h_v^{\mathcal{G},\theta}} \mathcal{L}(h_v^{\mathcal{G},\theta_s})^\top \mathbf{J}_{h_v^{\mathcal{G},\theta}\theta_s}(\theta - \theta_s)$$
$$= \frac{1}{2}(\theta - \theta_s)^\top \mathbf{J}^\top_{h_v^{\mathcal{G},\theta}\theta_s} \nabla^2_{h_v^{\mathcal{G},\theta}} \mathcal{L}(h_v^{\mathcal{G},\theta_s}) \mathbf{J}_{h_v^{\mathcal{G},\theta}\theta_s}(\theta - \theta_s). \tag{29}$$

Since $\theta^*_{\text{lin},\epsilon}$ is the optimal parameter minimizing the objective, the derivative of the objective with respect to $\theta$ at $\theta = \theta^*_{\text{lin},\epsilon}$ is zero:

$$0 = -\frac{1}{N} \sum_{v \in \mathcal{V}_{\text{train}}} \mathbf{J}^\top_{h_v^{\mathcal{G},\theta}\theta_s} \nabla^2_{h_v^{\mathcal{G},\theta}} \mathcal{L}(h_v^{\mathcal{G},\theta_s}) \mathbf{J}_{h_v^{\mathcal{G},\theta}\theta_s}(\theta^*_{\text{lin},\epsilon} - \theta_s) + \lambda(\theta^*_{\text{lin},\epsilon} - \theta_s)$$
$$+ \sum_{v \in \mathcal{V}_{\text{train}}} \epsilon \left(\nabla_\theta \mathcal{L}(h_v^{\mathcal{G},\theta_s}) - \nabla_\theta \mathcal{L}(h_v^{\mathcal{G}-\frac{1}{N},\theta_s})\right). \tag{30}$$

Rearranging the terms, we obtain:

$$\theta^*_{\text{lin},\epsilon} = \theta_s + \epsilon\, \mathbf{G}^{-1} \sum_{v \in \mathcal{V}_{\text{train}}} \left(\nabla_\theta \mathcal{L}(h_v^{\mathcal{G},\theta_s}) - \nabla_\theta \mathcal{L}(h_v^{\mathcal{G}-\frac{1}{N},\theta_s})\right), \tag{31}$$

where $\mathbf{G} := \frac{1}{N} \sum_{v \in \mathcal{V}_{\text{train}}} \mathbf{J}^\top_{h_v^{\mathcal{G},\theta}\theta_s} \nabla^2_{h_v^{\mathcal{G},\theta}} \mathcal{L}(h_v^{\mathcal{G},\theta_s}) \mathbf{J}_{h_v^{\mathcal{G},\theta}\theta_s} + \lambda\mathbf{I}$ is the generalized Gauss-Newton matrix.

Taking the derivative with respect to $\epsilon$, we obtain:

$$\frac{\partial \theta^*_{\text{lin},\epsilon}}{\partial \epsilon} = \lim_{\epsilon \to 0} \frac{\theta^*_{\text{lin},\epsilon} - \theta_s}{\epsilon} = \mathbf{G}^{-1} \sum_{v \in \mathcal{V}_{\text{train}}} \left(\nabla_\theta \mathcal{L}(h_v^{\mathcal{G},\theta_s}) - \nabla_\theta \mathcal{L}(h_v^{\mathcal{G}-\frac{1}{N},\theta_s})\right), \tag{32}$$

which confirms that the linearized edge-edit PBRF objective yields the same influence function as derived in Equation (10).

# C   Influence function for multiple edge edits

**Problem setup for multiple edge edits**   In this section, we extend our influence function to handle multiple edge edits. Let $\mathcal{S}$ denote the set of edges to be edited. We generalize the definition of $A^\epsilon$ in Section 3 to the multi-edge case by defining

$$A_{uv}^\epsilon = A_{uv} + \big(2\mathbb{I}[\{u,v\} \in \mathcal{S}] - 1\big)N\epsilon.$$

The edge-reweighted graph is then given by $\mathcal{G}^\epsilon = \{\mathcal{V}, \mathcal{E}, A^\epsilon\}$, where edges in $\mathcal{S}$ are reweighted. Setting $\epsilon = -1/N$ implies that all edges in $\mathcal{S}$ are deleted if they exist, and inserted otherwise.

**Parameter shift**   Following Section 3, the edge-edit PBRF is defined as

$$\theta_\epsilon^* := \arg\min_\theta \frac{1}{N} \sum_{v \in \mathcal{V}_{\text{train}}} D_{\mathcal{L}}\left(h_v^{\mathcal{G},\theta}, h_v^{\mathcal{G},\theta_s}\right) + \frac{\lambda}{2}\|\theta - \theta_s\|^2 + \sum_{v \in \mathcal{V}_{\text{train}}} \epsilon\left(\mathcal{L}\left(h_v^{\mathcal{G},\theta}\right) - \mathcal{L}\left(h_v^{\mathcal{G}^{-\frac{1}{N}},\theta}\right)\right). \tag{33}$$

Following the same derivation steps as in Appendix B, the contribution of the parameter shift to the evaluation function is given by

$$\nabla_\theta f(\theta_0^*, \mathcal{G})^\top \left.\frac{\partial \theta_\epsilon^*}{\partial \epsilon}\right|_{\epsilon=0} = -\nabla_\theta f(\theta_s, \mathcal{G})^\top \mathbf{G}^{-1} \sum_{v \in \mathcal{V}_{\text{train}}} \left(\nabla_\theta \mathcal{L}\left(h_v^{\mathcal{G},\theta_s}\right) - \nabla_\theta \mathcal{L}\left(h_v^{\mathcal{G}^{-\frac{1}{N}},\theta_s}\right)\right), \tag{34}$$

where $\mathbf{G} = \mathbf{J}_{h\theta_s}^\top \mathbf{H}_{h_s} \mathbf{J}_{h\theta_s} + \lambda\mathbf{I}$. The only distinction from the single-edge case lies in the redefinition of $\mathcal{G}^{-1/N}$, which now denotes the graph with multiple edge edits.

**Message propagation**   Unlike the single-edge case in Section 3, the perturbed adjacency matrix $A^\epsilon$ is influenced simultaneously by all edges in the set $\mathcal{S}$. Consequently, the message propagation effect can no longer be attributed to a single edge but must be decomposed into contributions from all edited edges. By applying the chain rule, the overall effect is expressed as a summation over gradients with respect to each reweighted edge:

$$\frac{\partial f(\theta, \mathcal{G}^\epsilon)}{\partial A^\epsilon}\frac{\partial A^\epsilon}{\partial \epsilon}\bigg|_{\theta=\theta_0^*, \, \epsilon=0} = \sum_{\{u,v\}\in\mathcal{S}} \frac{\partial f(\theta, \mathcal{G}^\epsilon)}{\partial A_{uv}^\epsilon}\frac{\partial A_{uv}^\epsilon}{\partial \epsilon}\bigg|_{\theta=\theta_s, \epsilon=0} + \frac{\partial f(\theta, \mathcal{G}^\epsilon)}{\partial A_{vu}^\epsilon}\frac{\partial A_{vu}^\epsilon}{\partial \epsilon}\bigg|_{\theta=\theta_s, \epsilon=0}$$

$$= \sum_{\{u,v\}\in\mathcal{S}} (2\mathbb{I}[\{u,v\} \in \mathcal{E}] - 1)N\left(\frac{\partial f(\theta_s, \mathcal{G})}{\partial A_{uv}} + \frac{\partial f(\theta_s, \mathcal{G})}{\partial A_{vu}}\right). \tag{35}$$

**Unified influence function under multiple edge edits**   By substituting the parameter shift and message propagation terms into Equation (8) and linearizing around $\epsilon = 0$, the resulting influence under multiple edge edits can be approximated as

$$f\left(\theta_{-\frac{1}{N}}^*, \mathcal{G}^{-\frac{1}{N}}\right) - f(\theta_s, \mathcal{G}) \approx \frac{1}{N}\nabla_\theta f(\theta_s, \mathcal{G})^\top \mathbf{G}^{-1} \sum_{v \in \mathcal{V}_{\text{train}}} \left(\nabla_\theta \mathcal{L}\left(h_v^{\mathcal{G},\theta_s}\right) - \nabla_\theta \mathcal{L}\left(h_v^{\mathcal{G}^{-\frac{1}{N}},\theta_s}\right)\right)$$

$$- \sum_{\{u,v\}\in\mathcal{S}} (2\mathbb{I}[\{u,v\} \in \mathcal{E}] - 1)\left(\frac{\partial f(\theta_s, \mathcal{G})}{\partial A_{uv}} + \frac{\partial f(\theta_s, \mathcal{G})}{\partial A_{vu}}\right). \tag{36}$$

# D    Description of LiSSA

To approximate the inverse of the generalized Gauss–Newton Hessian-vector product $\mathbf{G}^{-1}v$, we employ the LiSSA algorithm [1]. LiSSA estimates $\mathbf{G}^{-1}v$ by iteratively accumulating powers of the residual matrix $(\mathbf{I} - \mathbf{G})$ applied to the vector $v$. When the spectral radius of $(\mathbf{I} - \mathbf{G})$ is less than 1, the inverse can be expressed using the Neumann series:

$$\mathbf{G}^{-1}v = \sum_{k=0}^{\infty}(\mathbf{I} - \mathbf{G})^k v. \tag{37}$$

Letting $r^{(K)} = \sum_{k=0}^{K}(\mathbf{I} - \mathbf{G})^k v$, the iteration is defined recursively as:

$$r^{(0)} = v, \quad r^{(k+1)} = v + (\mathbf{I} - \mathbf{G})r^{(k)}. \tag{38}$$

In practice, we perform the update in Equation (38) until convergence. The iteration is terminated early if the update difference $\|r^{(k+1)} - r^{(k)}\|$ falls below a predefined threshold, or when the number of iterations reaches 10,000.

Since the spectral radius of $(\mathbf{I} - \mathbf{G})$ is not necessarily less than 1, we rescale the matrix to ensure convergence. Specifically, we define a scaled matrix $\mathbf{G}_s = \frac{1}{s}\mathbf{G}$, where $s > \lambda_{\max}(\mathbf{G})$, so that the spectral radius of $(\mathbf{I} - \mathbf{G}_s)$ is less than 1. The inverse is then computed via

$$\mathbf{G}^{-1}v = \frac{1}{s}\mathbf{G}_s^{-1}v,$$

and LiSSA is applied to approximate $\mathbf{G}_s^{-1}v$.

To avoid explicitly storing the generalized Gauss–Newton matrix $\mathbf{G}$, the matrix-vector product $\mathbf{G}r^{(k)}$ in Equation (38) is computed approximately using a Jacobian-based heuristic. Specifically, we compute the Jacobian-vector product $\boldsymbol{x} = \mathbf{J}_{h\theta_s}r^{(k)}$, apply the Hessian to obtain $\boldsymbol{x} \leftarrow \mathbf{H}_{h_s}\boldsymbol{x}$, and finally compute the transposed Jacobian-vector product $\mathbf{G}r^{(k)} = \mathbf{J}_{h\theta_s}^{\top}\boldsymbol{x}$.

# E    Experimental result on other datasets and GNNs

In this section, we present scatter plots for additional non-convex GNNs and datasets. Figure 7 shows the results for additional datasets, while Figure 8 and Figure 9 show the results for ChebNet [13] and GAT [33], respectively. Under these settings, our influence function accurately predicts the actual influence, consistently showing a correlation above 0.8.

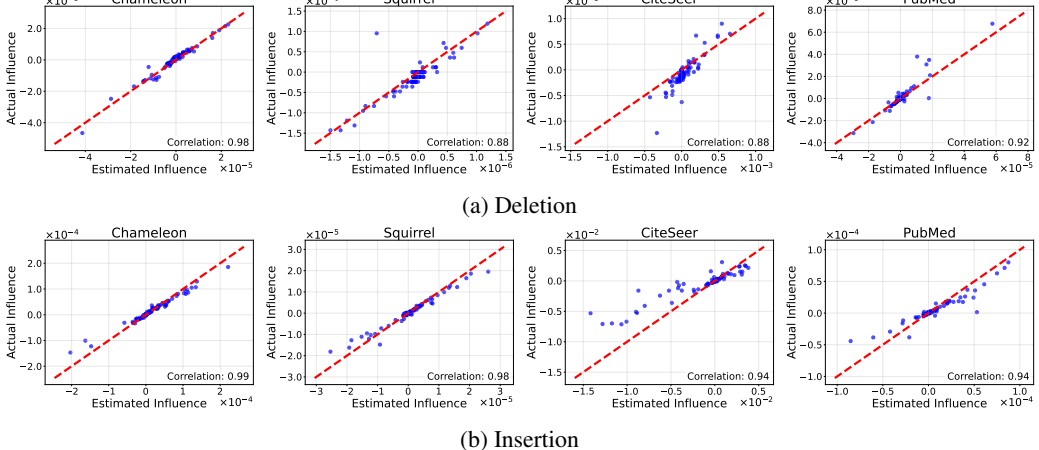

(a) Deletion

(b) Insertion

Figure 7: The scatter plot of predicted influence and actual influence on four-layer GCN. The x-axis represents the predicted influence and y-axis represents the actual influence, and the red-dotted line represents the perfect alignment.

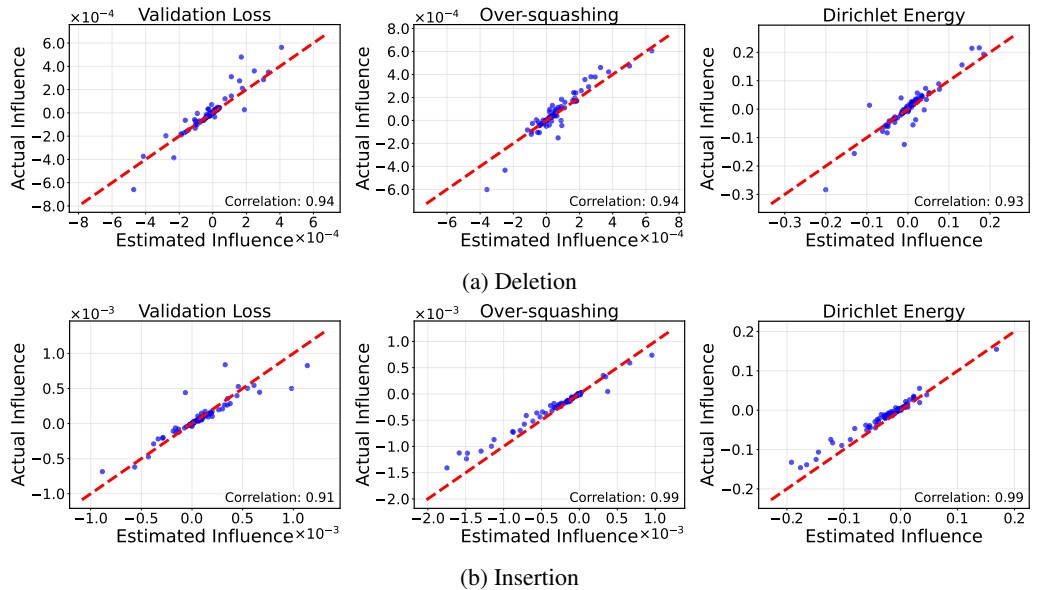

(a) Deletion

(b) Insertion

Figure 8: The scatter plot of predicted influence and actual influence on two-layer ChebNet. The x-axis represents the predicted influence and y-axis represents the actual influence, and the red-dotted line represents the perfect alignment.

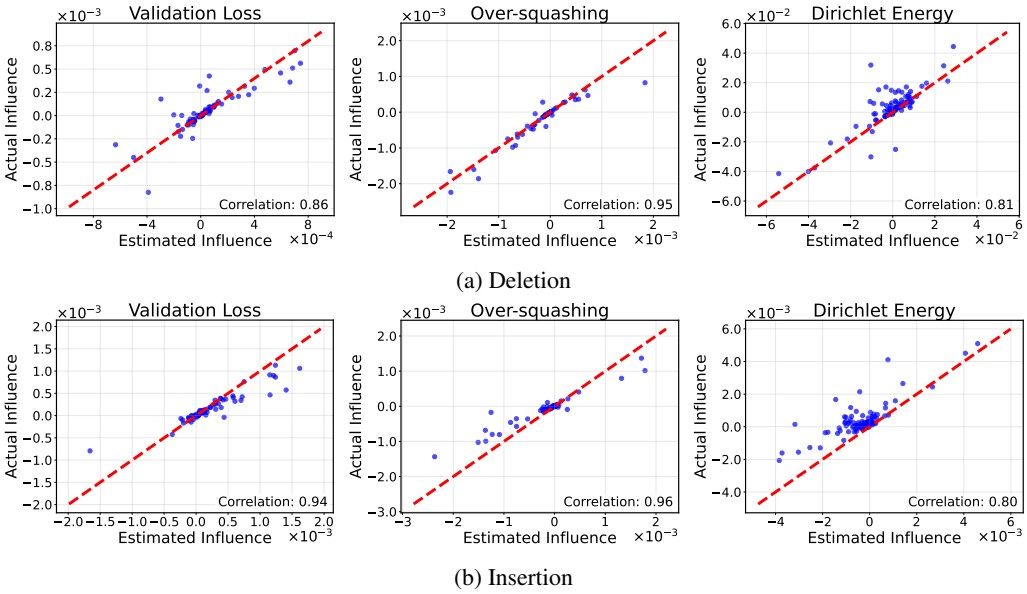

(a) Deletion

(b) Insertion

Figure 9: The scatter plot of predicted influence and actual influence on two-layer GAT. The x-axis represents the predicted influence and y-axis represents the actual influence, and the red-dotted line represents the perfect alignment.

## F   Experimental configuration

All experiments are conducted using NVIDIA GeForce RTX 3090, NVIDIA RTX A5000, and NVIDIA RTX A6000 GPUs. The experiments presented in the main text employ a 4-layer GCN model as a representative non-convex GNN. To produce the results in Table 1 and Table 2, we tune the model and training hyperparameters of a vanilla GCN over the following search space: learning rates $\{0.1, 0.03, 0.01\}$, hidden dimensions $\{32, 64\}$, and weight decays $\{10^{-3}, 10^{-4}, 10^{-5}, 10^{-6}, 10^{-7}\}$. Training is performed for 2000 epochs using the SGD optimizer. For influence function computation,

we run the LiSSA algorithm for 10,000 iterations. The damping parameter $\lambda$ is selected from $\{0.1, 0.01, 0.001, 0.0001\}$. We randomly sample 10,000 candidate edges for both deletion and insertion, and estimate their influence. For Table 1, the number of edges to edit is determined based on validation accuracy. The experiments are repeated for 10 independent runs using different random seeds provided by the BernNet implementation.

**BORF** [26] and **FoSR** [19], used in the analysis of edge rewiring methods, are applied with the default settings from the original paper. The number of rewired edges reported in Figure 6 is aggregated over 10 runs, with a total of 200 edges inserted. Unlike FoSR, which performs only edge insertions, BORF also considers edge deletions. To improve visualization, we reduce the number of edge deletions, as the influence scores for insertions exhibit a long-tailed distribution, making it difficult to display them on the same scale as deletions. Accordingly, edge deletions are performed with 120, 100, and 100 edges, respectively.

For the adversarial attack experiments in Table 2, **DICE** [34] and **PRBCD** [14] are implemented by modifying the PyTorch-based DeepRobust library [24], while maintaining its default settings. For our method, we consider both edge insertions and deletions, and select the operations with the highest influence scores for validation loss, and the lowest influence scores for over-squashing and Dirichlet energy.

