# OpenReview forum: "Influence Functions for Edge Edits in Non-Convex Graph Neural Networks"
_NeurIPS.cc/2025/Conference — NeurIPS 2025 poster_

### Official Review · Reviewer_q9wj · 2025-06-16

**Clarity:** 3
**Significance:** 2
**Originality:** 3
**Rating:** 4
**Confidence:** 3

**Summary:**

The paper proposes a novel influence-function framework based on an approximate Proximal Bregman response function for edge edits in non-convex graph neural networks (GNNs), covering both edge deletions and insertions. It decomposes the effect of an edge edit into parameter-shift and message-propagation components. Extensive experiments on multiple datasets and GNN architectures show that predicted effects correlate highly with actual effects on validation loss, over-squashing, and Dirichlet energy, significantly outperforming previous GIF methods. Finally, the paper demonstrates practical applications of this influence function in analyzing graph rewiring strategies, constructing adversarial attacks, and studying the impact of homophilic versus heterophilic edges

**Questions:**

1. This paper only selects one baseline to compare, which is not convincing enough. Since there are other related baselines, such as [1].
2. Why does the legend in the Figures refer to Dirichlet Energy instead of the term "over-smoothing" used in the main text? Could the authors clarify the reasoning behind this terminology choice and whether these two are being used interchangeably or have distinct implications in the analysis?
3. Does assigning values of |ϵ| other than 1/N carry any practical or theoretical significance in this study? Can the authors elaborate on whether other values of |ϵ| were considered, and if so, what their implications would be for influence estimation accuracy or interpretability?
4. In Section 4.1, is the definition of an "L-layer neighborhood" based on shortest path distance? The term "L-layer" is introduced in Section 4.1 when defining neighborhood influence. For clarity, could the authors confirm whether this refers to the set of nodes within shortest-path distance L from the target node, or if another graph-theoretic interpretation is used?

[1] Gong L, Cheng Q. Exploiting edge features for graph neural networks[C]//Proceedings of the IEEE/CVF conference on computer vision and pattern recognition. 2019: 9211-9219.

**Ethical Concerns:**

["NO or VERY MINOR ethics concerns only"]

**Final Justification:**

The specified details of experiments and generalization ability in practical scenarios are discussed in the authors' response.

**Limitations:**

Yes

**Quality:**

3

**Strengths And Weaknesses:**

Strengths
1. This paper introduces an influence-function framework based on the PBRF, which relaxes convexity assumptions and, for the first time, systematically supports both edge deletions and insertions in non-convex GNNs.
2. Decomposes the impact of an edge edit into parameter shift and message propagation components, and empirically demonstrates their low correlation.
3. This paper conducts extensive experiments across multiple benchmark datasets and diverse GNN architectures, further validates practical value via downstream graph editing for improved accuracy, analysis of rewiring strategies, adversarial attack construction, and homophilic/heterophilic edge impact studies.

Weaknesses
1. The experiment is not convincing enough. This paper only selects one baseline to compare, however, there are other related baselines, such as [1].
2. The paper proposes using the LiSSA algorithm to approximate the inverse Hessian–vector product, but does not specify any details regarding the number of iterations, step-size selection, or convergence guarantees. Furthermore, there is no quantitative assessment of how the LiSSA approximation error propagates into the final influence estimation.
3. The current framework addresses only single-edge edits. In practical scenarios, multiple edges may be modified simultaneously, leading to potentially significant nonlinear coupling effects.
4. The influence function derivation relies solely on a first-order Taylor approximation and does not account for higher-order or cross-term effects. The manuscript should either provide a theoretical justification for truncating at first order, demonstrating that higher-order contributions are indeed negligible, or include an empirical analysis quantifying the error introduced by ignoring these terms.
5. The last term in Equation (9), which measures the difference in node-level loss between the original and edge-edited graphs, is presented in a form that is difficult to interpret.

[1] Gong L, Cheng Q. Exploiting edge features for graph neural networks[C]//Proceedings of the IEEE/CVF conference on computer vision and pattern recognition. 2019: 9211-9219.

---

> ### Author Rebuttal · Authors · 2025-07-30
>
> We sincerely thank the reviewer for their thorough and constructive feedback. We particularly appreciate the reviewer recognizing the contributions of our influence-function framework, including relaxing convexity assumptions, handling both edge deletions and insertions, and conducting extensive empirical validations across various scenarios. We have carefully read all the comments and have addressed each concern and question in detail below. If there are any further questions or suggestions, we would be happy to respond.
>
> **[W1, Q1] The experiment is not convincing enough. This paper only selects one baseline to compare, however, there are other related baselines, such as [1].**
>
> We sincerely appreciate the suggestion regarding additional baselines. We have carefully reviewed the paper suggested by the reviewer. Our understanding is that the objective of [1] is to propose a GNN architecture designed to effectively exploit multi-dimensional edge features. We think this objective differs from that of our paper, which introduces an influence function tailored specifically for node classification tasks. If there is a connection or relevance between [1] and our work that we have overlooked, we would sincerely appreciate it if the reviewer could let us know.
>
> To the best of our knowledge, the only existing baseline directly addressing influence functions on graphs is GIF [2, 3], which we have extensively compared against in our paper.
>
> **[W2] The paper proposes using the LiSSA algorithm, but does not specify any details regarding the number of iterations, step-size selection, or convergence guarantees. Furthermore, there is no assessment of how the LiSSA approximation error propagates into the final influence estimation.**
>
> - **[Hyperparameter details for LiSSA]** We respectfully refer the reviewer to Appendix F (lines 509–510), where we describe the hyperparameters used for the LiSSA algorithm. Specifically, we run LiSSA for $10,000$ iterations, and select the damping parameter $\lambda$ from the set $\\{0.1, 0.01, 0.001, 0.0001\\}$.
> - **[LiSSA convergence details]** Details on how we ensure the convergence of the LiSSA algorithm are provided in Appendix C. In brief, convergence holds when the spectral radius of $\mathbf{I}-\mathbf{G}$ is less than 1. To enforce this, we scale the matrix as $\mathbf{G}_s=\frac{1}{s}\mathbf{G}$ and increment $s$ when divergence is observed. We respectfully refer the reviewer to Appendix C for a full description.
> - **[Quantitative analysis on LiSSA approximation errors]** To quantitatively assess how the LiSSA approximation error propagates into the final influence estimation, we measured the Pearson correlation between the estimated influence (via LiSSA) and the actual influence (via retraining) across varying LiSSA iteration counts. The results are summarized below:
>
> |LiSSA_iter|Cora|CiteSeer|PubMed|
> |-|-|-|-|
> |10|-0.16|-0.04|0.76|
> |100|0.16|0.04|0.93|
> |500|0.91|0.94|0.94|
> |1000|0.91|0.94|0.94|
> |10000|0.91|0.94|0.94|
> |Time per iteration|11.8ms|18.0ms|19.5ms|
>
> These results demonstrate that low iteration counts (10 or 100) produce poor influence estimates due to high approximation errors. However, with 500 iterations or more, LiSSA provides accurate and stable influence estimations.
>
> **[W3] The current framework addresses only single-edge edits. In practical scenarios, multiple edges may be modified simultaneously, leading to potentially significant nonlinear coupling effects.**
>
> We agree with the reviewer that addressing multiple-edge edits is practically important and worth clarifying. Our framework can be extended to the multiple-edge setting with only a minor change to Equation (12).
> - **[Retraining term modification]** We redefine $h_v^{\mathcal{G}^{-1/N}}$ as the representation of node $v$ in the graph where all selected edges have been edited.
> - **[Message passing term modification]** In the second term of Equation (12), we introduce a summation over the perturbed edge set, i.e., we prepend $\sum_{(u,v)}$ to account for every edited edge.
>
> We evaluate the Pearson correlation between the actual influence (obtained by retraining) and our estimated influence when simultaneously inserting $k$ edges on the Cora dataset:
> |k|1|5|10|20|50|100|1000|
> |-|-|-|-|-|-|-|-|
> |Correlation|0.91|0.88|0.91|0.93|0.88|0.86|0.79|
>
> The results demonstrate that our influence estimation remains accurate across a broad range of simultaneous edge edits. A minor decrease in correlation is observed only at very high levels of edge editing ($k = 1,000$), which we attribute to first-order approximation errors arising from significant parameter changes when editing a large number of edges.
>
> **[W4] The influence function derivation relies solely on a first-order Taylor approximation and does not account for higher-order or cross-term effects. The manuscript should either provide a theoretical justification for truncating at first order, or include an empirical analysis quantifying the error introduced by ignoring these terms.**
>
> We adopt the first-order Taylor approximation primarily due to the significant computational complexity associated with higher-order approximations. To illustrate the inherent difficulty, we consider the case of the vanilla influence function [4] as a simpler example (since our specific influence function is even more complex).
>
> For clarity, we include the second-order derivative $\frac{d^2\hat{\theta}_{z,\epsilon}}{d\epsilon^2}$ at $\epsilon=0$ below:
>
> $$
> -H_{\hat{\theta}}^{-1}\left(\nabla_{\theta}^{3}R(\hat{\theta})\left[-H_{\hat{\theta}}^{-1}\nabla_{\theta}\mathcal{L}(z,\hat{\theta}), -H_{\hat{\theta}}^{-1}\nabla_{\theta}\mathcal{L}(z,\hat{\theta})\right]-2\nabla_{\theta}^2\mathcal{L}(z,\hat{\theta})H_{\hat{\theta}}^{-1}\mathcal{L}(z,\hat{\theta})\right)
> $$
>
> Computing this second-order derivative involves (1) evaluating the third-order derivative of the total training loss $R(\theta)=\sum_{v\in V_{\text{train}}}\mathcal{L}(v,\theta)$and (2) computing the second-order derivative of the loss for the specific data point $z$ whose influence is being assessed. These terms are challenging to compute, as evaluating higher-order derivatives with respect to the parameters typically requires substantial computational time and memory resources. Despite the truncation at the first-order approximation, our empirical evaluations indicate that this approximation yields high correlation (up to 0.95) with the actual influence, confirming its practical adequacy.
>
> **[W5] The last term in Equation (9), which measures the difference in node-level loss between the original and edge-edited graphs, is presented in a form that is difficult to interpret.**
>
> To enhance the interpretability of Equation (9), we will explicitly clarify the meaning of the term $h_v^{\mathcal{G}^{-1/N,\theta}}=\text{GNN}(\{\mathcal{V},\mathcal{E}/\{e\},\mathbf{X}\},\theta)$ which denotes the node embedding after removing the edge $e$, directly following Equation (9). We believe this clarification will enhance readers' understanding.
>
> **[Q2] Why does the legend in the Figures refer to Dirichlet Energy instead of the term "over-smoothing" used in the main text? Could the authors clarify the reasoning behind this terminology choice?**
>
> Dirichlet Energy is a widely adopted metric from prior research for quantifying over-smoothing [5,6]. It directly measures the average difference between representations of adjacent nodes, intuitively capturing the degree of over-smoothing (lower Dirichlet Energy indicates stronger over-smoothing). We respectfully direct the reviewer to Section 4.1 for additional information.
>
> **[Q3] Does assigning values of |ϵ| other than 1/N carry any practical or theoretical significance in this study? Can the authors elaborate on whether other values of |ϵ| were considered, and if so, what their implications would be for influence estimation accuracy or interpretability?**
>
> The parameter $|\epsilon|$ controls the magnitude of an edge edit: when $|\epsilon| = 1/N$ (with $N$ denoting the total number of nodes), it represents fully deleting or adding an edge, whereas $|\epsilon| < 1/N$ corresponds to partially reducing or increasing an edge’s influence. Since our primary goal is to measure the influence of fully inserting or removing an edge, we consistently set $|\epsilon| = 1/N$ throughout our study. This choice is standard in influence function research, where the focus is generally on measuring the impact of completely removing or adding an individual data point (or edge, in our context) [2,4]. Exploring smaller values of $|\epsilon|$ could potentially provide insights into partial modifications of edges, but this falls beyond the main scope of our current analysis.
>
> **[Q4] In Section 4.1, is the definition of an "L-layer neighborhood" based on shortest path distance? For clarity, could the authors confirm whether this refers to the set of nodes within shortest-path distance L from the target node, or if another graph-theoretic interpretation is used?**
>
> We carefully revisited our manuscript based on the reviewer's comment, but we were unable to locate the term "$L$-layer neighborhood." Perhaps the reviewer was referring to the "$L$-hop neighborhood," which is the term we used in Section 4.1. If this is the case, we would like to clarify that an "$L$-hop neighborhood" refers to the set of nodes whose shortest-path distance to the target node is at most $L$.
>
> [1] Gong, Liyu, et al. "Exploiting edge features for graph neural networks." CVPR 19.
> [2] Chen, Zizhang, et al. "Characterizing the influence of graph elements." ICLR 23.
> [3] Wu, Jiancan, et al. "Gif: A general graph unlearning strategy via influence function." ACM 23.
> [4] Koh, Pang Wei, et al. "Understanding black-box predictions via influence functions." ICML 17.
> [5] Cai, Chen, et al. "A note on over-smoothing for graph neural networks." arXiv 20.
> [6] Rusch, T, et al. "A survey on oversmoothing in graph neural networks." arXiv 23.

---

> > ### Comment · Reviewer_q9wj · 2025-08-06
> > **Official Comment by Reviewer q9wj**
> >
> > Thanks for your detailed response! I am considering raising the score.

---

> > > ### Author Response · Authors · 2025-08-06
> > >
> > > We are grateful for the response and the kind consideration. Many thanks once again for taking the time to provide thoughtful feedback.

---

### Official Review · Reviewer_XMsy · 2025-07-02

**Clarity:** 4
**Significance:** 3
**Originality:** 3
**Rating:** 5
**Confidence:** 3

**Summary:**

This paper, in general, presents their own method of predicting the edge influence in GNNs. What's so interesting about their work, in my opinion, is that they greatly relaxed the assumption we put on the GNNs to be analyzed, from convex to non-convex.

Their method can evaluate the influence of edges towards many different "objectives", such as measuring towards the over-squashing, over-smoothing, or the standard validation loss (such as cross-entropy). They are able to detect some "beneficial" edges that contributes to the phenomenon, and some "harmful" edges that avoid or break such phenomenon.

The capability of their method is also shown in downstream experiments such as adversarial attacks.

**Questions:**

1. I have some questions on the exact scalability and efficiency of your method. If, for example, I am planning to apply such method on a very large graph, say, social network graphs that typically include at least tens of thousands of nodes, do you think it will be problematic?

2. The downstream tasks you proposed are interesting, but I really want to see if there are more applications. Say, do you think your method can be useful in some graph-based causal analysis studies, given that you relaxed the assumptions to non-convex graphs?

**Ethical Concerns:**

["NO or VERY MINOR ethics concerns only"]

**Final Justification:**

The authors addressed all my concerns with care, and beyond. The quality of their work is slightly higher than I expected, so I am willing to increase my score a little bit.

**Limitations:**

yes

**Quality:**

3

**Strengths And Weaknesses:**

Their work has very clearly explained theoretical proof, and provided justification in all decisions they made. Preciseness is their greatest strength. The work is carefully-done in general, with obvious novelty, and the message is clearly delivered.
And also, they provided their code and the code is well-documented. I think that helps explain a lot of the implementation details that are probably not sufficiently discussed in the main text.

The weaknesses are relatively subtle, but still there.
Although the technical part is carefully written, and roughly justified, we did not see much mathematical proof on some of the decisions, such as that the alternative over-squashing measurement can really replace the a gradient-based metric.

Also, some of the wordings really worth re-considering, such as that "beneficial" and "harmful" are really counter-intuitive. You will understand their meaning after carefully reading their paper though.

All in all, I think this paper is above the acceptance line, but the writing (and proofing) can be improved somehow.

---

> ### Author Rebuttal · Authors · 2025-07-30
>
> We sincerely thank the reviewer for their thorough and positive review, as well as their encouraging comments on the clarity, novelty, and careful implementation of our work. We particularly appreciate the recognition of our detailed theoretical explanations and well-documented code. Below, we address the concerns and valuable suggestions raised, and we look forward to any further feedback from the reviewer.
>
> **[W1] We did not see much mathematical proof on some of the decisions, such as that the alternative over-squashing measurement can really replace the a gradient-based metric.**
>
> We acknowledge the reviewer's concern regarding the lack of a mathematical proof for our alternative over-squashing measurement. While we agree that a rigorous mathematical justification demonstrating the replaceability of the two metrics would be ideal, conducting such a formal analysis is not straightforward. As an alternative, we provide empirical evidence demonstrating that our proposed metric is highly correlated with the gradient-based measure and significantly more computationally efficient. For every node in each dataset, we computed both metrics, reporting the Pearson correlation coefficients and runtimes below:
>
> ||Cora|CiteSeer|PubMed|
> |-|-|-|-|
> |Grad runtime|27.6s|26.0s|614.8s|
> |Ours runtime|3.6s|3.9s|27.6s|
> |Correlation|0.74|0.87|0.67|
>
> Our alternative measure achieves strong correlations (0.67 to 0.87) with the gradient-based metric while offering substantial runtime improvements (6 to 20 times faster). We emphasize that these two metrics measure the same underlying concept, quantifying how much a node’s representation changes when the features of the most distant node in its receptive field are perturbed.
>
> **[W2] Some of the wordings really worth re-considering, such as that "beneficial" and "harmful" are really counter-intuitive.**
>
> We agree with the reviewer that the terms "beneficial" edge and "harmful" edge can be counter-intuitive. To address this issue, we will revise our terminology to explicitly reflect both the operation performed (removal/insertion) and its impact on the evaluation metric, adopting terms such as "beneficial edge removal" or "metric-increasing edge removal." We believe this revised terminology will resolve two key sources of confusion: (1) it explicitly clarifies that the terms refer specifically to the effect observed after an edge operation, and (2) it avoids the confusion caused by the opposite direction of impact between the edge's original role and the operation being performed. These changes should intuitively clarify the relationship between edge edits and their effects on the evaluation metric.
>
>
> **[Q1] I have some questions on the exact scalability and efficiency of your method. If, for example, I am planning to apply such method on a very large graph, say, social network graphs that typically include at least tens of thousands of nodes, do you think it will be problematic?**
>
> We do not anticipate significant issues applying our method to large graphs containing tens of thousands of nodes. In fact, our experiments include evaluations on the PubMed dataset [1], which contains 19,717 nodes. We provide the GPU memory consumption and running time on this dataset:
>
> - **[Used GPU]** NVIDIA A6000 (48GB)
> - **[GPU memory consumption]** 1,386MB
> - **[Running time]** 41.04 seconds for calculating the influence of 1,000 edge edits (LiSSA calculation: 15.28 seconds, Remaining calculation: 25.76 seconds).
>
> These results confirm that our method is practical and efficient for graphs at this scale, demonstrating clear applicability to large-scale scenarios.
>
> Furthermore, our method can also scale to very large graphs by employing mini-batch training, which divides the original graph into ten subgraphs following the approach used in [2]. We provide the GPU memory consumption, running time, and Pearson correlation on ogbn-products (2,449,029 nodes) [3]:
>
> - **[Used GPU]** NVIDIA A6000 (48GB)
> - **[GPU memory consumption]** 31,700 MB
> - **[Running time]**
>   - LiSSA preprocessing: 258.3 seconds
>   - Influence function calculation per edge (excluding LiSSA preprocessing): 26.4 seconds
>   - Note: LiSSA preprocessing is required only once, after which calculating the influence function for multiple edges significantly reduces overhead (total average 26.7 seconds per edge when computing 1000 edges).
> - **[Pearson correlation]** 0.94
>
>
> **[Q2] The downstream tasks you proposed are interesting, but I really want to see if there are more applications. Say, do you think your method can be useful in some graph-based causal analysis studies, given that you relaxed the assumptions to non-convex graphs?**
>
> Yes, our influence function can be applied to various tasks involving non-convex graph neural networks, including graph-based causal analysis. Specifically, by setting the evaluation function as the loss associated with a single node, our method can identify which edges most strongly influence the model's prediction for that node [4]. Therefore, it is well-suited for causal analyses aimed at understanding the impact of specific edges on node-level predictions. However, as we are not experts in causality, a more rigorous and formal analysis would be required to validate this application thoroughly. We believe this represents an exciting direction for future research.
>
> Moreover, by selecting a fairness metric as the evaluation function, our framework can help identify edges whose removal would enhance fairness in model predictions [5]. Additionally, since our approach quantifies the extent of model parameter changes due to edge edits, it is applicable to model unlearning scenarios that require the removal of specific relationships [6]. We believe these applications highlight exciting opportunities for further research leveraging our method.
>
> [1] Sen, Prithviraj, et al. "Collective classification in network data." AI magazine 29.3 2008.
> [2] Li, Guohao, et al. "Deepergcn: All you need to train deeper gcns." arXiv preprint arXiv 2020.
> [3] Hu, Weihua, et al. "Open graph benchmark: Datasets for machine learning on graphs." NeurIPS 2020.
> [4] Koh, Pang Wei, and Percy Liang. "Understanding black-box predictions via influence functions." ICML 2017.
> [5] Chhabra, Anshuman, et al. "" What Data Benefits My Classifier?" Enhancing Model Performance and Interpretability through Influence-Based Data Selection." ICLR 2024.
> [6] Zhang, Yang, et al. "Recommendation unlearning via influence function." ACM Transactions on Recommender Systems 2024.

---

### Official Review · Reviewer_ofas · 2025-07-02

**Clarity:** 3
**Significance:** 3
**Originality:** 3
**Rating:** 4
**Confidence:** 3

**Summary:**

This work proposes a method for estimating the influence functions of edge edits in graphs. Specifically, the authors introduce a graph-tailored proximal Bregman response function that supports both edge insertion and deletion while accounting for their effects on parameter shifts and message propagation. Experiments on three graph datasets demonstrate that the proposed method effectively estimates influence functions. By selectively adding or removing edges based on these estimates, the method can either enhance predictive performance or degrade it in adversarial scenarios.

**Questions:**

Do the changes in the evaluation function caused by modifications in the message propagation path still persist if only the training graph is edited, but the test graph remains unchanged?

**Ethical Concerns:**

["NO or VERY MINOR ethics concerns only"]

**Final Justification:**

My concerns have been addressed, and I maintained my positive evaluation.

**Limitations:**

Yes

**Quality:**

3

**Strengths And Weaknesses:**

Strengths:

1. Estimating the influence function of edge edits is a valuable and underexplored problem with promising practical applications.

2. The paper proposes a practical and effective solution for estimating influence functions tailored to graph data. This is supported by experimental evaluations, which show a small error between the estimated influence scores and ground truth.

3. The proposed method is backed by solid theoretical analysis.

Weaknesses:

1. The method is evaluated on three datasets—Cora, Citeseer, and PubMed—which are all homophilous graphs, where most edges connect nodes of the same class. Since the paper also investigates the influence of inserting or deleting homophilous versus heterophilous edges, it would be more convincing and comprehensive to include evaluations on heterophilous graphs or graphs with lower homophily.

2. In Table 1, the results are based on models trained and evaluated on edge-edited graphs. It is unclear whether the comparisons remain fair if the test graphs are edited in different ways across methods. Clarifying this setup would improve the rigor of the evaluation. Also, is it appropriate to edit the test graph?

3. The improvements reported in Table 1 do not appear to be statistically significant compared to the baselines. All methods show relatively similar performance, making it difficult to draw strong conclusions about how over-squashing or over-smoothing affects model performance. This may be due to the chosen datasets being relatively insensitive to edge edits. Again, incorporating a more diverse set of datasets could help better assess the method’s impact.

---

> ### Author Rebuttal · Authors · 2025-07-30
>
> We sincerely thank the reviewer for their thorough review and valuable feedback on our submission. We especially appreciate the reviewer recognizing the theoretical rigor, and practical effectiveness of our method for estimating influence functions in graph data. Below, we provide detailed responses addressing the weaknesses and questions raised. We hope our responses effectively clarify these points and look forward to any additional comments from the reviewer.
>
> **[W1] The method is evaluated on three datasets—Cora, Citeseer, and PubMed—which are all homophilous graphs, where most edges connect nodes of the same class. Since the paper also investigates the influence of inserting or deleting homophilous versus heterophilous edges, it would be more convincing and comprehensive to include evaluations on heterophilous graphs or graphs with lower homophily.**
>
> We report the correlation coefficients and scatter plots for two heterophilic datasets in Figure 6 of Appendix D. The correlations range from 0.88 to 0.99, confirming that our influence function remains accurate even under lower homophily. Due to the inability to embed images directly in the rebuttal, we respectfully direct the reviewer to Appendix D for detailed plots.
>
> Additionally, we evaluated test accuracy after editing edges selected to reduce validation loss on these heterophilic graphs. The summarized results in the table below suggest that our edge-editing method for reducing validation loss also effectively improves test accuracy on heterophilic graphs:
>
> ||Chameleon|Actor|
> |-|-|-|
> |GCN|52.2±3.9|32.4±2.1|
> |+Ours(VL)|53.3±4.2|34.9±1.4|
>
> **[W3] The improvements reported in Table 1 do not appear to be statistically significant compared to the baselines. All methods show relatively similar performance, making it difficult to draw strong conclusions about how over-squashing or over-smoothing affects model performance. This may be due to the chosen datasets being relatively insensitive to edge edits. Again, incorporating a more diverse set of datasets could help better assess the method’s impact.**
>
> We acknowledge the reviewer’s point regarding the modest improvements observed in Table 1, and agree these results alone provide limited evidence for our claims about over-smoothing and over-squashing.
>
> To further substantiate our claim, we direct the reviewer’s attention to Table 2 in our paper, which presents adversarial edge edits selected using our influence function. The variant designed to maximize validation loss consistently results in a significant accuracy drop: 2.6% on Cora, 4.2% on CiteSeer, and 5.7% on PubMed. In contrast, the variants targeting the over-smoothing and over-squashing metrics induce substantially smaller or negligible performance degradation. This clear difference supports our original claim that adversely affecting these structural properties does not necessarily translate into impaired predictive performance.
>
> Moreover, we provide additional evaluation results on two more diverse datasets (Chameleon and Actor) in response to the reviewer's earlier point ([W1]). These datasets further confirm the effectiveness and general applicability of our method.
>
>
> **[W2, Q1] In Table 1, the results are based on models trained and evaluated on edge-edited graphs. It is unclear whether the comparisons remain fair if the test graphs are edited in different ways across methods. Clarifying this setup would improve the rigor of the evaluation. Also, is it appropriate to edit the test graph? Do the changes in the evaluation function caused by modifications in the message propagation path still persist if only the training graph is edited, but the test graph remains unchanged?**
>
> We believe the comparisons in Table 1 remain fair due to the transductive node classification setup used in our paper. In transductive node classification, there is only one graph, and thus the training and test graphs are not separate. Instead, nodes within the graph are divided into training, validation, and test sets. During training, only training-node features are provided, along with all edges, including those connected to validation or test nodes. This setting is widely adopted in node classification [1,2,3]. Our method edits the edges provided at the training stage, which we believe ensures fairness. This follows the same protocol as previous edge-rewiring methods [4,5]. Consequently, perturbations in the message-propagation path affect both training and test nodes.
>
>
> [1] Kipf, Thomas N., and Max Welling. "Semi-supervised classification with graph convolutional networks." ICLR 2017.
> [2] Veličković, Petar, et al. "Graph attention networks." ICLR 2018.
> [3] Chien, Eli, et al. "Adaptive universal generalized pagerank graph neural network." ICLR 2021.
> [4] Nguyen, Khang, et al. "Revisiting over-smoothing and over-squashing using ollivier-ricci curvature." ICML 2023.
> [5] Karhadkar, Kedar, Pradeep Kr Banerjee, and Guido Montúfar. "FoSR: First-order spectral rewiring for addressing oversquashing in GNNs." ICLR 2023.

---

### Official Review · Reviewer_CafE · 2025-07-02

**Clarity:** 3
**Significance:** 3
**Originality:** 3
**Rating:** 5
**Confidence:** 4

**Summary:**

This paper builds one new kind of influence function for graph neural networks (GNNs). Old influence work needs the loss to be strictly convex and can only tell what happens if we delete one edge. The authors drop the convex need and let the formula work for deep, non-convex GNNs. They also treat both deleting and adding an edge as the same small change on edge weight.

Their “edge-edit proximal Bregman response” splits the total effect of one edge into two parts: (1) how model parameters move, (2) how message passing route changes. They use LiSSA to solve the inverse Hessian so the cost is low.

On five citation graphs and three GNN models, the predicted influence has about 0.9 Pearson link with the real effect, much better than past methods. They show three uses: cleaning or re-wiring graphs to improve accuracy, finding worst edges to attack the model fast, and explaining which edges help or hurt.

**Questions:**

1. Scale proof – Please run on one large OGB graph (e.g., ogbn-products). Report wall-time, peak GPU RAM, and Pearson r.

2. Stronger baselines – Add recent influence / attack tools (e.g., NID-Edge, I-F) and PGD multi-edge attack.

3. Multi-edge effect – Real edits often change dozens of edges. Can your first-order formula compose influences linearly? Please test on random k-edge batches (k = 5, 10, 20).

4. Reproducibility tips – Provide heuristic for choosing LiSSA depth and damping λ based on dataset size and model depth.

**Ethical Concerns:**

["NO or VERY MINOR ethics concerns only"]

**Final Justification:**

The authors show a large-graph case (ogbn-products) with subgraph mini-batch, report clear wall-time, peak GPU, and Pearson=0.94. Ablations are helpful: LiSSA depth plateaus around 500 iterations; removing propagation term hurts a lot. Multi-edge extension is explained and tested.

**Limitations:**

1. Scale risk. Need one large OGB graph case with wall-time and GPU RAM; without this, “scalable” claim is weak.

2. Baseline gap. Please compare with newer influence / attack tools such as NID-Edge, I-F, and multi-edge PGD.

3. Multi-edge reality. Real users edit many edges at once. Show whether first-order scores add up linearly for k = 5-20 edges.

4. Reproducibility. Give a thumb-rule for LiSSA depth and damping λ that changes with |V|, |E|, and model depth. This will help others copy the method.

**Quality:**

3

**Strengths And Weaknesses:**

Strengths

1. Authors drop the strict-convexity need and design one unified influence rule that works for deep, non-convex GNN and treats edge delete + add in the same frame. This fresh idea fills a clear blank in influence study.

2. On five citation graphs and three GNN types, predicted-vs-real influence hits ≈ 0.9 Pearson, and three demos (graph repair, attack, explain) show real help; LiSSA keeps cost low.

Weakness

1. Quality & Scope – Paper studies only first-order, single-edge change. Real tasks often edit many edges; LiSSA inverse still heavy on huge graphs (e.g., OGB). Thus “scalable” claim not fully proved.

2. Limited baselines and ablation – The study mainly compares with GIF and simple gradient scores. It lacks newer influence/attack tools such as NID or I-F. Also, no ablation on LiSSA depth, Hessian-free vs. exact inverse, or removing the propagation term, so we cannot see which design really drives the gains.

---

> ### Author Rebuttal · Authors · 2025-07-30
>
> We sincerely thank the reviewer for their thorough review and valuable feedback on our submission. We particularly appreciate the reviewer highlighting the novelty of our unified influence function and its strong empirical validation. Below, we carefully address the concerns regarding scalability, baseline comparisons, and ablations. We hope our responses effectively address these points and warmly welcome any further feedback from the reviewer.
>
> **[W1, Q1, L1] LiSSA inverse still heavy on huge graphs (e.g., OGB). Thus “scalable” claim not fully proved. Please run on one large OGB graph (e.g., ogbn-products). Report wall-time, peak GPU RAM, and Pearson r.**
>
> We would like to clarify that we do not claim scalability in our manuscript. In fact, we state in our conclusion that future research could focus to enhance the method’s applicability to larger and more complex graph learning tasks. Nevertheless, we are grateful for the reviewer's insightful comment regarding scalability, and following this suggestion, we have conducted additional experiments on the large-scale ogbn-products dataset (with 2,449,029 nodes) to empirically verify the applicability of our method to large graphs.
>
> Due to memory constraints inherent in GNNs when handling such large-scale graphs, we employed mini-batch training by dividing the original graph into ten subgraphs, following the approach used in [1]. Below, we report detailed experimental results:
>
> - **[GPU used]** NVIDIA RTX A6000 (48 GB memory)
> - **[Wall-time]**
>   - LiSSA preprocessing: 258.3 seconds
>   - Influence function calculation per edge (excluding LiSSA preprocessing): 26.4 seconds
>   - Note: LiSSA preprocessing is required only once, after which calculating the influence function for multiple edges significantly reduces overhead (total average 26.7 seconds per edge when computing 1000 edges).
> - **[Peak GPU memory consumption]** 31,700 MB
> - **[Pearson correlation]** 0.94
>
> These results suggest that our proposed method can be extended to large-scale graphs.
>
> **[W2, Q2, L2] The study mainly compares with GIF and simple gradient scores. It lacks newer influence/attack tools such as NID, I-F and PGD multi-edge attack.**
>
> Regarding the concern about comparison with newer influence/attack methods, we have now included PGD [2] in our adversarial-attack comparisons using the same perturbation budgets (1%, 3%, 5%) as those used in the main paper. The updated results are shown below:
>
> ||Cora(81.0)|||CiteSeer(69.3)|||PubMed(75.6)|||
> |-|-|-|-|-|-|-|-|-|-|
> |Attack|1%|3%|5%|1%|3%|5%|1%|3%|5%|
> |PGD|80.9±0.4|80.8±0.4|80.5±0.9|69.2±0.4|69.0±0.5|69.0±0.5|75.5±1.0|75.2±0.9|75.1±1.0|
> |PRBCD|80.6±0.5|79.4±0.5|78.7±0.9|68.5±0.5|67.4±0.8|66.4±0.5|74.6±1.5|72.9±1.8|70.9±1.7|
> |Ours(VL)|**80.2±0.7**|**79.1±0.7**|**78.4±0.8**|**68.2±0.5**|**66.7±0.6**|**65.1±0.9**|**73.2±1.4**|**71.2±1.5**|**69.9±1.1**|
>
> Our influence-function-based method consistently outperforms PGD across all datasets and perturbation budgets. We emphasize that adversarial attacks represent only one possible application of our influence functions, and even with our simple strategy of selecting the top-$k$ edges by influence function magnitude, our method consistently outperforms the compared baselines. This finding aligns with the results from PRBCD [3], a gradient-based attack method similar to PGD.
>
> We were unable to identify the methods "NID" and "I-F" mentioned by the reviewer. We would appreciate it if the reviewer could provide references for these methods.
>
> **[W2] No ablation on LiSSA depth, Hessian-free vs. exact inverse, or removing the propagation term, so we cannot see which design really drives the gains.**
>
> Regarding the ablations requested, we conducted experiments on LiSSA depth and the propagation term. Due to computational constraints, performing an ablation with the exact inverse of the Hessian (which has dimensions $p\times p,$ with $p$ being the number of parameters) was infeasible. Thus, we could not include a comparison with the exact inverse.
>
> - **[Ablation on LiSSA Depth]**
>
> We measured the Pearson correlation between the estimated influence and the actual influence (obtained via retraining) across varying numbers of LiSSA iterations. The results are summarized in the following table:
>
> |LiSSA iterations|Cora|CiteSeer|PubMed|
> |-|-|-|-|
> |10|-0.16|-0.04|0.76|
> |100|0.16|0.04|0.93|
> |500|0.91|0.94|0.94|
> |1000|0.91|0.94|0.94|
> |10000|0.91|0.94|0.94|
> |Time per iteration|11.8ms|18.0ms|19.5ms|
>
> These results indicate that the correlation plateaus at around 500 iterations, suggesting that approximately 500 iterations are sufficient to obtain accurate influence estimates. Additionally, each LiSSA iteration is computationally inexpensive (around 10s total for 500 iterations), making this overhead negligible.
>
> - **[Ablation on Propagation Term]**
>
> We measured the Pearson correlation between the estimated influence and the actual influence after removing the propagation term. Results are shown below:
>
> ||Cora|CiteSeer|PubMed|
> |-|-|-|-|
> |w/o propagation|0.00|0.45|0.69|
> |Ours|0.91|0.94|0.94|
>
> Removing the propagation term causes a significant drop in correlation, clearly demonstrating that this term is essential for accurately estimating the influence function.
>
> **[W1, Q3, L3] Real edits often change dozens of edges. Can your first-order formula compose influences linearly? Please test on random k-edge batches (k = 5, 10, 20).**
>
> Unlike influence functions for non-relational data, graph-based influence functions typically are non-additive, meaning that summing single-edge influences does not directly yield the influence of multiple edge edits [4]. Nevertheless, our formulation naturally extends to multi-edge edits with minimal modifications to Equation (12):
> - **[Retraining term modification]** We redefine $h_v^{\mathcal{G}^{-1/N}}$ as the node representation after all selected edges are edited.
> - **[Message passing term modification]** We modify Equation (12) by summing over all perturbed edges, i.e., we introduce $\sum_{(u,v)}$ in the second term to account for multiple edge edits.
>
> To empirically validate this approach, we measured the Pearson correlation between the actual influence and our estimated influence across different sizes of randomly inserted edges (inserting $k$ edges simultaneously) on the Cora dataset:
> |k|1|5|10|20|50|100|1000|
> |-|-|-|-|-|-|-|-|
> |Correlation|0.91|0.88|0.91|0.93|0.88|0.86|0.79|
>
> These results demonstrate that our influence estimates remain highly accurate for a wide range of $k$. Only at a very large scale $k = 1,000$ does the correlation drop slightly, which we attribute to first-order approximation errors caused by significant parameter changes from editing a large number of edges simultaneously.
>
> **[Q4, L4] Reproducibility tips – Provide heuristic for choosing LiSSA depth and damping λ based on dataset size and model depth.**
>
> - **[LiSSA depth]** We use a convergence-based heuristic, where at each iteration, we compute the $l_2$-norm of the difference between the current and previous inverse-Hessian estimates. The iteration stops when this difference falls below $10^{-7}$. We set the maximum depth to $10,000$.
> - **[Damping $\lambda$]** A fixed damping value of $\lambda=0.01$ consistently provided strong performance across nearly all evaluated datasets.
>
>
> [1] Li, Guohao, et al. "Deepergcn: All you need to train deeper gcns." arXiv preprint arXiv 2020.
> [2] Xu, Kaidi, et al. "Topology attack and defense for graph neural networks: An optimization perspective." IJCAI 2019.
> [3] Geisler, Simon, et al. "Robustness of graph neural networks at scale." NeurIPS 2021.
> [4] Song, Jaeyun, SungYub Kim, and Eunho Yang. "Rge: A repulsive graph rectification for node classification via influence." ICML 2023.

---

### Note · Authors · 2025-08-13

We deeply appreciate all the valuable time and dedicated efforts spent reviewing our submission. In this final remark, we summarize the strengths and weaknesses highlighted by the reviewers, along with our responses addressing each of these concerns.

### **Strengths**
- **Novelty (CafE, q9wj, ofas, XMsy)**
Reviewers mentioned novelty as a key strength of our work: extending the influence function to non-convex GNNs and enabling it to handle both edge deletions and insertions.
- **Clarity of Theoretical Proofs (ofas, XMsy)**
Reviewers highlighted that our work provides clearly presented theoretical proofs.
- **Strong Empirical Results (CafE, ofas, XMsy)**
Reviewers noted the strength of our empirical results, which show a 0.9 correlation with actual influence.
- **Interesting Practical Applications (CafE, q9wj, ofas, XMsy)**
Reviewers emphasized the value of the applications demonstrated in our work, such as the analysis of rewiring strategies, adversarial attacks, and studies on homophilic and heterophilic edges.
- **Other Strengths**
One reviewer highlighted the thorough documentation of our code **(XMsy)**, and another reviewer mentioned the computational efficiency of LiSSA **(CafE)**.

### **Weaknesses and Our Responses**
- **Limited to Single Edge Edit (CafE, q9wj)&rarr;Demonstrating Capability on Multiple Edge Edits**
Reviewers noted the necessity of estimating multiple edge edits. In response, we demonstrated our method's capability on up to 1,000 simultaneous edge edits, achieving correlations of 0.79–0.93 with actual influence.
- **Scalability to Large-scale Graphs (CafE, XMsy)&rarr;Empirical Demonstration on Large-scale Graphs**
Reviewers raised concerns about scalability to large-scale graphs. In response, we provided empirical results on ogbn-products (2.4M nodes), showing that our method calculates the influence within 26.7 seconds per edge on a single A6000 GPU.
- **Limited Ablations (CafE, q9wj)&rarr;Conducting Additional Ablation Studies**
Reviewers requested additional ablation studies. In response, we conducted experiments confirming that the chosen LiSSA depth is sufficient and that the propagation term is essential for accurate estimation.
- **Other Weaknesses**
To address other weaknesses, we provided comparisons against PGD attacks **(CafE)**, evaluations on heterophilic datasets **(ofas)**, empirical validation of our proposed metric **(XMsy)**, and heuristics for LiSSA hyperparameter selection **(CafE)**.

---

### Decision · Program_Chairs · 2025-09-17

**Decision:**

Accept (poster)

**Comment:**

The paper studies edge influence functions in GNNs without relying on the convexity assumption of the loss function. It introduces a proximal Bregman response function tailored to graphs, which can accommodate both edge additions and deletions. The reviewers identified several merits, including the novel problem formulation targeting non-convex GNNs, the sound theoretical analysis, and the supporting empirical evaluation. The concerns raised during the review process have been reasonably addressed in the rebuttal. Overall, the paper makes a well-motivated contribution to the literature with interesting practical applications to graph rewiring and adversarial attacks.